# Enhancing Extreme Weather Forecasting via Dynamically Weighted MSE

## Abstract

Data-driven weather forecasting empowered by deep learning has shown superior performance compared to traditional physics-based dynamical models. However, conventional training objectives (like Root Mean Squared Error (RMSE)) primarily focus on minimizing average prediction errors, often resulting in oversmoothed forecasts that fail to capture critical extreme weather phenomena, including heavy precipitation, hurricanes, and other high-impact events. To overcome this limitation, we propose a robust loss function, named Dynamically Weighted MSE (DW-MSE), that adaptively reweights training samples to better learning on extreme weather events. Specifically, we introduce a dual-branch meta-network alongside the prediction network to dynamically generate sample weights: one branch captures spatiotemporal dependencies across climate variables, while the other learns from training losses. Guided by a small set of validation samples, the meta-network can be jointly optimized with the prediction network via an efficient bi-level optimization strategy, which provides the fast convergence with the approximated first-order information. Overall, our framework is able to accurately identify and assign greater importance to extreme weather samples without manually designing reweight function and any prior knowledge. Extensive experiments in both training-from-scratch and fine-tuning climate models demonstrate that DW-MSE consistently outperforms existing approaches in forecasting extreme weather.

## 1 Introduction

Weather forecasting (Bi et al., 2023; Liu et al., 2025; Verma et al., 2024; Ling et al., 2024; Chen et al., 2025a; Gao et al., 2025) plays an essential role in society, influencing a wide range of industries and daily activities, such as transportation and agriculture. Accurate forecasts are particularly crucial for predicting extreme weather events, which significantly impact both human lives and economic stability. For example, *Hurricane Ian* in 2022 caused over $112 billion in damages in the United States, making it one of the costliest hurricanes in history (Bucci et al., 2022). These examples highlight the urgent need for robust weather forecasting models, especially for the accuracy and lead time of extreme weather event predictions, to ensure effective disaster prevention and response.

Recently, deep (or machine) learning-based approaches have emerged as a promising alternative for weather forecasting (Zhao et al., 2025; Gong et al., 2025; Xu et al., 2024b; Chen et al., 2024; 2025b). By leveraging large datasets and powerful neural network architectures, these methods bypass the need for explicit physical modeling and instead learn intricate patterns and relationships directly from data. For example, Pangu-Weather (Bi et al., 2023) has shown superior performance in predicting temperature, wind speed, wind direction, and barometric pressure compared to traditional NWP models like the European Centre for Medium-Range Weather Forecasts (ECMWF) (Owens & Hewson, 2018). Instead of the prediction accuracy, deep learning approaches also exhibit advantages in that they can process high-dimensional inputs efficiently and are particularly adept at capturing complex nonlinear data (Xu et al., 2024a).

However, deep learning-based weather models are predominantly trained using regression loss functions, such as (Root) Mean Squared Error (RMSE/MSE) (Xu et al., 2024a; Verma et al., 2024; Chen et al., 2023; Subich et al., 2025), which often leads to poor performance in forecasting extreme weather events. These loss functions primarily focus on minimizing overall prediction error, fre-

quently at the cost of accurately capturing rare but high-impact events (Xu et al., 2024a). As a result, such models may perform suboptimally in disaster warning scenarios, including the prediction of hurricanes, heavy rainfall, or heatwaves. This limitation is particularly critical, as accurate forecasting of extreme weather is essential for reducing the severe societal and economic consequences associated with these events.

Designing a robust loss function that encourages the model to better capture extreme values or distributions has emerged as a promising approach (Lopez-Gomez et al., 2023; Xu et al., 2024a; Ni, 2023). These loss functions typically rely on manually defined weighting schemes—such as exponential adjustments (Lopez-Gomez et al., 2023) or fixed weights (Xu et al., 2024a)—that assign different levels of importance to normal and extreme weather values. For instance, a recent work (Xu et al., 2024a) proposed an asymmetric variant of the mean squared error loss, known as Exloss, which applies higher weights to prediction errors involving extreme events. This design boosts the influence of extreme samples (e.g., those in the top or bottom 10th percentile) during model training, thereby enhancing robustness. However, the manual specification of weighting functions, as seen in (Xu et al., 2024a; Meo et al., 2024), heavily depends on prior knowledge. This often leads to rigid and inflexible weight assignments, ultimately limiting the scalability and adaptability of such methods.

Motivated by MW-Net (Shu et al., 2019), a meta learning-based sample reweighting approach originally developed for noisy-label learning (Shu et al., 2019; Ren et al., 2018), we propose to dynamically reweight climate events to improve performance in extreme weather forecasting. However, directly applying this method faces two major limitations: (1) the limited capacity of the reweighting network, which struggles to capture the complexity of high-dimensional, time-dependent climate data, and (2) the high training cost incurred by the bi-level optimization process, involving the computation of second-order Hessian matrices.

To address these limitations, we propose a novel framework, dubbed dynamically weighted MSE (DW-MSE), that automatically learns sample weights for the prediction model to better fit extreme distributions while ensuring lower training costs. Specifically, we design a dual-branch reweight network alongside the prediction model, which replaces manually crafted weighting functions in previous methods with an adaptive mechanism. One branch leverages training dynamics (e.g., loss information) to identify hard or extreme samples, while the other captures spatiotemporal dependencies directly from raw climate fields. Based on a carefully constructed validation set, these two networks can be jointly optimized by an efficient bi-level optimization strategy, which is structured into two steps: 1) *Inner loop:* generate variable- and location-aware weights for training samples and update the prediction network with our weighted MSE loss. 2) *Outer loop:* optimize the reweight network on a validation set containing only extreme events, using first-order information for efficiency. Through this design, DW-MSE enables the prediction network to up-weight rare yet high-impact climate samples without relying on manually specified weighting functions.

In summary, our contribution can be summarized into three aspects:

- We propose DW-MSE, a meta-learning framework that adaptively reweights training samples via a dual-branch reweight network, which captures training dynamics and spatiotemporal dependencies for accurately weight assignment.

- We design an efficient bi-level optimization algorithm that avoids expensive second-order computations by leveraging approximate first-order information, ensuring computational efficiency.

- Extensive experiments in multiple climate forecasting settings verify that DW-MSE consistently improves the prediction of extreme events, while maintaining strong overall forecasting performance and outperforming existing loss functions.

## 2 RELATED WORK

**Weather forecasting.** Previously, weather forecasting has been widely studied as a sub-task of the field of Time-Series Forecasting (Zhou et al., 2021; Wu et al., 2021), which is identical to other time-series tasks like Traffic Flow Prediction (Ji et al., 2023; Li et al., 2021), Stock Price Prediction (Zhang et al., 2024; Qin et al., 2024), and Energy Load Forecasting (Zhou et al., 2021). Due to the simplicity of the adapted network and the low quality of the data, these methods struggle to support large-scale weather prediction as the number of climate variables and geographic locations

increases. Recently, with the improvement of computation resources and the release of high-quality climate data, many organizations like Google (Lam et al., 2022), Microsoft (Nguyen et al., 2023), and Huawei (Bi et al., 2023) published their weather foundation models trained on data at the scale of tens of terabytes (TB). The key points within these deep-learning-based approaches lie in the advancing network. Although data-driven models outperform the physical-based ECMWF-IFS (Owens & Hewson, 2018) in standard metrics such as RMSE, they often produce overly smooth predictions and underperform in extreme events.

**Extreme weather forecasting.** Extreme weather forecasting has attracted significant attention due to its critical real-world applications in early warning of natural disasters, such as the prediction of hurricanes, severe storms and heavy rainfall to mitigate potential damage and save lives (Ran et al., 2024). Methods in this field are generally categorized into two types: 1) network design-based methods and 2) robust loss design-based methods. In this paper, we focus solely on the design of the loss function, as it can be seamlessly integrated into any network to enhance performance in predicting extreme weather values.

In the family of *robust loss functions*, existing methods normally adapt the MSE loss by defining a loss weighted function to assign larger weights for the extreme cases. For example, (Lopez-Gomez et al., 2023) proposes a custom loss function based on an exponential L2 distance for extreme samples, assigning different fixed weights to positive and negative extremes. Exloss (Xu et al., 2024a) employs an asymmetric scaling function to allocate weights for extreme samples, and EVloss (Meo et al., 2024) proposes an exponential adjustment function to gradually increase the weight for extreme cases. Despite non-trival improvements, the design of weighted functions in these methods heavily relies on prior knowledge (like extreme distributions) and thus suffers from limited scalability. By contrast, our method automatically learns a set of weights for all training samples without access to any prior knowledge, offering strong adaptability.

**Meta learning,** also known as "learning to learn", enables rapid model adaptation for new tasks by leveraging prior knowledge and optimizing learning strategies (Finn et al., 2017; Nichol, 2018; Snell et al., 2017). Inspired by the performance of meta-learning, some methods were proposed to automatically learn hyper-parameters (Real et al., 2020; Jaderberg et al., 2017), dynamically adjust and search network architecture (Zoph et al., 2018; Liu et al., 2019), reweight training samples (Shu et al., 2019; Ren et al., 2018), and sample/core-set selection (Yang et al., 2022).

Compared with previous methods, the challenge underlying the learning weight for weather data lies in the intricate interdependencies among climate variables (also called uncertainty in other works (Xu et al., 2024a)), which often exhibit non-linear and dynamic relationships across varying spatial and temporal scales. This complexity imposes higher demands on meta-learning, as the learned weights have to not only capture these multifaceted relationships but also adapt effectively to diverse scenarios and extreme weather conditions, ensuring robust and accurate predictions.

## 3 PRELINIMARIES

**Problem Definition.** We consider the data-driven climate forecasting task under the setting of fixed-size input and output windows. The training dataset is defined as $\mathcal{D}^{\text{train}} = \{(X_n, Y_n)\}_{n \in [N]}$, where $X_n = \{\mathbf{x}_1, ..., \mathbf{x}_t\}_n$ denotes the input climate sequence of length $t$, and $Y = \{\mathbf{y}_{t+1}, ..., \mathbf{y}_{t+U}\}_n$ denotes the corresponding target sequence of length $U$. Note that $\mathbf{x}, \mathbf{y} \in \mathbb{R}^{H \times W \times V}$ denotes a meteorological field consisting of $V$ variables on a spatial grid of size $H \times W$.

Given a model $f_{\boldsymbol{\theta}}(\cdot)$ parameterized by $\boldsymbol{\theta}$, the prediction for the input sequence $n$ is $\hat{Y}_n = f_{\boldsymbol{\theta}}(X_n)$. The model is trained by minimizing the average discrepancy between predictions and ground truth over all samples, which is defined as

$$\mathcal{L}^{\text{train}} = \frac{1}{N} \sum_{n=1}^{N} L(Y_n, \hat{Y}_n), \tag{1}$$

where $L(\cdot)$ denotes the loss function, regularly the Root Mean Squared Error (MSE) (Xu et al., 2024a), which can be written as $L(Y, \hat{Y}) = \sqrt{\frac{1}{UHWV} \sum_{u=1}^{U} \sum_{h=1}^{H} \sum_{w=1}^{W} \sum_{v=1}^{V} (\mathbf{y}_{u;h,w,v}, \hat{\mathbf{y}}_{u;h,w,v})^2}$. Note that we omit an altitude-related coefficient for convenience.

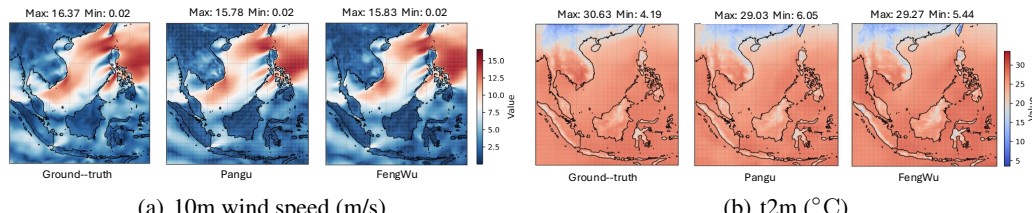

(a) 10m wind speed (m/s)  (b) t2m (°C)

Figure 1: **Smooth prediction issue**. We visualize the ground truth and predictions from two climate models (Pangu and FengWu) for *2m_temperature* and *10m wind speed*. Despite strong overall performance, both models produce smooth results, limiting their usefulness for disaster early warning.

## 3.1 CHALLENGE: SMOOTH PREDICTIONS IN EXTREME WEATHER FORECASTING

Although current data-driven weather models have achieved performance comparable to or even surpassing that of physics-based forecasting systems, they often struggle with predicting extreme weather events (Nguyen et al., 2023). Specifically, they tend to produce overly smooth outputs, failing to capture sharp, localized variations and instead generating non-extreme values. This limitation hinders their effectiveness in disaster early warning, where accurate and timely predictions of extremes are critical.

Figure 1 visualizes the predictions of two representative models - Pangu (Bi et al., 2023) and ClimaX (Nguyen et al., 2023) - compared to ground truth. We observe that these models often underestimate localized extremes due to excessive smoothing. Notably, this issue persists across different models, variables, and forecasting horizons, highlighting its widespread impact on current climate modeling techniques.

Here, an intuitive explanation for this issue is given from the perspective of the loss function. To minimize the overall error, MSE/RMSE inherently favors global error averaging, leading the model to fit common weather patterns (i.e., high-probability "normal" values) while neglecting the accurate prediction of extreme weather events. Currently, some works propose the robust loss function to mitigate this issue, such as EXloss (Xu et al., 2024a) and EVLoss (Meo et al., 2024), which leverage the loss re-weighting strategy by assigning larger weights to extreme weather events and smaller weights to regular events. Despite nontrival improvements, their weight assignment mechanism heavily relies on prior knowledge and designing the adjustment function manually. This manner largely restricts their approach's flexibility and scalability, especially when adapting or fine-tuning to another area's weather conditions.

## 4 METHODOLOGY

To avoid manually defining the weight-adjust function for MSE losses, we propose to automatically learn weights across all climate variables and locations in a mete-learning manner, named dynamically weighted MSE (DW-MSE), which largely enhances the flexibility of the loss re-weighting framework while keeping lower training cost.

Concretely, while prior approaches assign a scalar weight $\lambda \in \mathbb{R}$ at the sample level (e.g., larger for extreme events and smaller for others), we refine this idea by learning a spatially-varying weight matrix $\lambda \in \mathbb{R}^{H \times W \times V}$ for each weather map, which allows more fine-grained control within a single sample during model learning. Compared to Eq. (1), the weighted training loss for the prediction network can be reformulated as

$$\mathcal{L}^{\text{train}} = \frac{1}{N} \sum_{n=1}^{N} \lambda_n \cdot L(Y_n, \hat{Y}_n). \tag{2}$$

Unlike previous approaches (Xu et al., 2024a; Lopez-Gomez et al., 2023), we automatically learn this weight through a carefully designed weighted-network $g_{\boldsymbol{\omega}}(\cdot)$, parameterized by $\boldsymbol{\omega}$, rather than relying on a manually defined adjustment function.

At this point, our framework consists of two networks, each influencing the other. Specifically, the generation quality of $\lambda$ influences the learning process of the prediction network, and a good

performance of the prediction network boosts the meta-network. To jointly optimize parameters of both two networks, we resort to a bi-level optimization framework for iterative improvement. The nested optimization objective [1] can be written as

$$\boldsymbol{\theta}^*(\boldsymbol{\omega}) = \arg\min_{\boldsymbol{\theta}} \mathcal{L}^{\text{train}}\big(Y^{\text{train}}, f_{\boldsymbol{\theta}}(X^{\text{train}}); \lambda^{\text{train}}(\boldsymbol{\omega})\big) \ \text{ s.t. } \arg\min_{\boldsymbol{\omega}} \ \mathcal{L}^{\text{val}}(Y^{\text{val}}, f_{\boldsymbol{\theta}^*(\boldsymbol{\omega})}(X^{\text{val}})). \quad (3)$$

The first term in Eq. (3) is prediction network learning based on training data re-weighted by the data weight model, while the second term is to train the weighted network on a manually constructed validation set which only contains the extreme events.

In what follows, we introduce 1) how to build a high-capability meta-net, 2) how to efficiently solve the nested optimization problem in Eq. (3), and 3) the construction of the validation set.

### 4.1 ARCHITECTURE OF THE WEIGHED NETWORK

We design the weighted network $g_{\boldsymbol{\omega}}$ as a dual-branch meta-net to address the limitations of using a single source of information for weighting. Specifically, relying solely on training losses (like MW-Net (Shu et al., 2019)) tends to assign larger weights to samples with higher errors, but it ignores the global information in weather maps, thereby missing extreme weather signals or rare-value patterns. On the other hand, raw weather data contain rich spatiotemporal structures that can highlight local–global climate patterns and extreme regions, but without considering the actual prediction errors, the network may misallocate weights to visually salient yet less impactful areas. By jointly learning from training losses and raw weather data, the proposed dual-branch design allows the meta-net to emphasize high-error samples while capturing physically

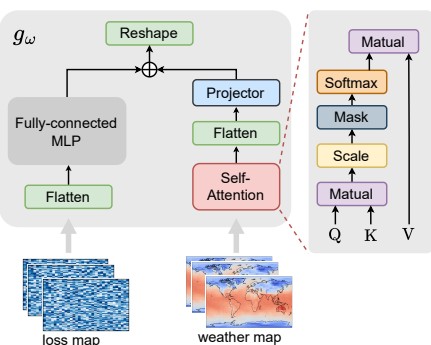

Figure 2: **Illustration of the weighted net**, which learns from both losses and raw weather data and then outputs the weights map.

meaningful spatial patterns, ultimately generating more reliable and fine-grained weight maps.

To be specific, 1) in the first branch of *learning from losses*, the input for the sample $X_n$ is the corresponding loss map, represented by $\ell(Y_n, f_{\boldsymbol{\theta}}(X_n)) \in \mathbb{R}^{H \times W \times V}$, which is flattened to a vector of one dimension and then passed through fully connected layers to obtain a compact feature representation. This branch ensures that regions with higher prediction errors (probably the underfitting extreme events) receive proportionally larger weights. 2) In the second branch of *learning from raw weather map*, we introduce a multi-head self-attention (MSA) structure (Voita et al., 2019) to enhance the network's ability to model the local and global weather values, contributing to the weighted net being aware of extreme cases. Finally, we take the outputs from the two branches and compute their average, which is then reshaped into the spatial grid. To guarantee that the weights fall within a valid range, we deploy a Sigmoid activation function to the last layer that constrains each value to $[0, 1]$.

In summary, for sample $n$, the output of this weighted-net can be represented by

$$\lambda_n = g_{\boldsymbol{\omega}}(X_n, \ell(Y_n, f_{\boldsymbol{\theta}}(X_n))) \in \mathbb{R}^{H \times W \times V}. \quad (4)$$

Figure 2 shows the architecture of our proposed meta-net. More details of the MLP architecture and the self-attention module are provided in the Appendix A.1.

### 4.2 EFFICIENT BI-LEVEL OPTIMIZATION

Obtaining the optimal parameters $\boldsymbol{\theta}^*$ and $\boldsymbol{\omega}^*$ (in Eq. (3)) is a nested optimization problem that can be solved by bi-level optimization (Shu et al., 2019; Finn et al., 2017). Specifically, we build an online optimization processes that iteratively update the parameters of two networks. In each iteration $t$, the parameters of two networks are denoted by $\boldsymbol{\theta}^{(t)}$ and $\boldsymbol{\omega}^{(t)}$, respectively. The iterative optimization process can be divided into the following two steps:

---

[1]The superscripts $\texttt{train}$ and $\texttt{val}$ indicate that the samples are drawn from the training set and the validation set, respectively.

- Prediction Model Update. At $t$-th iteration, we first update the prediction model via the reweighted training objective in Eq. (2). Specifically, given a mini-batch $D_b^{\text{train}}$ with the sample size of $b$, the weighted network generate the corresponding weight matrix $\lambda(\boldsymbol{\omega}^{(t)})$. The parameter update follows:

$$\boldsymbol{\theta}^{(t+1)} = \boldsymbol{\theta}^{(t)} - \alpha \nabla_{\boldsymbol{\theta}} \mathcal{L}^{\text{train}} \Big( Y^{\text{train}}, f_{\boldsymbol{\theta}^{(t)}}(X^{\text{train}}); \lambda^{\text{train}}(\boldsymbol{\omega}^{(t)}) \Big). \tag{5}$$

- Weighted Network Update. The weighted network is then updated based on the validation loss, which is depends the updated parameter $\boldsymbol{\theta}^{(t+1)}$:

$$\boldsymbol{\omega}^{(t+1)} = \boldsymbol{\omega}^{(t)} - \eta \nabla_{\boldsymbol{\omega}} \mathcal{L}^{\text{val}} \big( Y^{\text{val}}, f_{\boldsymbol{\theta}^{(t+1)}(\boldsymbol{\omega})}(X^{\text{val}}) \big). \tag{6}$$

where $\alpha$ and $\eta$ denotes the step size during optimization.

**Gradient Analysis.** In Eq. (6), the gradient of the validation loss *w.r.t.* $\boldsymbol{\omega}$ can be formulated as

$$\nabla_{\boldsymbol{\omega}} \mathcal{L}^{\text{val}} = \frac{\partial \mathcal{L}^{\text{val}}}{\partial f_{\boldsymbol{\theta}}} \cdot \frac{\partial f_{\boldsymbol{\theta}}}{\partial \boldsymbol{\theta}^{(t+1)}} \cdot \frac{\partial \boldsymbol{\theta}^{(t+1)}}{\partial \boldsymbol{\omega}}, \tag{7}$$

Since the parameter $\boldsymbol{\theta}^{(t+1)}$ in the last term is dependent on $\boldsymbol{\omega}$ thought Eq. (5), we have $\frac{\partial \boldsymbol{\theta}^{(t+1)}}{\partial \boldsymbol{\omega}} = -\alpha \frac{\partial^2 \mathcal{L}^{\text{train}}}{\partial \boldsymbol{\theta} \, \partial \boldsymbol{\omega}}$, where the Hessian vector product is introduced. Therefore, the original bi-level optimization makes the exact meta-gradient computationally expensive in practice, especially for high-dimensional climate prediction models.

To address this limitation, we refer to the first-order approximation strategy for MAML Finn et al. (2017), where the core-idea is to ignore the second-order derivative term (e.g., the last term in Eq. (7)). The meta-gradient can be approximated by treating $\boldsymbol{\theta}^{(t+1)}$ as fixed *w.r.t.* $\boldsymbol{\omega}$, which is written as $\nabla_{\boldsymbol{\omega}} \mathcal{L}^{\text{val}} = \frac{\partial \mathcal{L}^{\text{val}}}{\partial f_{\boldsymbol{\theta}}} \cdot \frac{\partial f_{\boldsymbol{\theta}}}{\partial \boldsymbol{\theta}^{(t+1)}}$. Accordingly, the parameter update for the weighted network reduces to

$$\boldsymbol{\omega}^{(t+1)} \approx \boldsymbol{\omega}^{(t)} - \eta \nabla_{\boldsymbol{\omega}} \mathcal{L}^{\text{val}} \big( Y^{\text{val}}, f_{\boldsymbol{\theta}^{(t+1)}}(X^{\text{val}}) \big). \tag{8}$$

Although the second-order information is discarded, the approximation still allows the weighted network to adapt effectively, as it captures the dominant influence of $\boldsymbol{\omega}$ through its direct effect on the reweighted training process. A more discussion about the effectiveness of the approximated meta-gradient can be found in the Appendix A.2. Through the iterative application of Eq. (5) and Eq. (8), the two networks can gradually tend toward convergence.

**Validation Set Construction.** To emphasize the importance of rare but critical events, we deliberately construct the validation set to contain only extreme climate samples. Specifically, for each weather map, we retain the top $\rho$ and bottom $\rho$ fraction of grid values corresponding to extreme highs and lows, while masking out the remaining non-extreme regions. This design ensures that the validation loss focuses exclusively on extreme events, thereby guiding the weighted network to assign higher importance to these regions. By doing so, the meta-optimization process avoids dilution from normal samples and explicitly encodes our objective of improving prediction accuracy under extreme weather conditions. In all experiments, this parameter $\rho$ is set as $10\%$.

## 5 EXPERIMENTS

### 5.1 EXPERIMENTAL DETAILS

**Datasets.** Following previous works (Nguyen et al., 2023; Meo et al., 2024; Xu et al., 2024a), we build the test benchmarks on the 5th generation of ECMWF reanalysis (ERA5) platform. Considering the popularity of climate foundation models, we construct two test settings, including 1) fine-tuning climate foundation models (like ClimaX) for regional forecasting using a small resolution of $5.625°$, and 2) training from scratch with a larger resolution of $1.40625°$. More details of these two benchmarks can be found in Appendix B.

**Evaluation Metrics.** We employ three metrics to evaluate the performance of different approaches - two focusing on overall accuracy and one dedicated to extreme weather forecasting. **1) Root mean square error (RMSE)**, which measures the average magnitude of prediction errors by taking the square root of the mean of squared differences between predicted and observed values. **2) Anomaly**

Table 1: Comparison results of RMSE (↓) on *regional cliamte forecasting*. We highlight the best performance with the **bold** font. Our proposal DW-MSE achieve the state-of-the-art performance under almost all settings.

| Variable | Hours | North-America | | | South-America | | | Australia | | |
|---|---|---|---|---|---|---|---|---|---|---|
| | | RMSE | EXloss | DW-MSE | RMSE | EXloss | DW-MSE | RMSE | EXloss | DW-MSE |
| z500 | 6 | 278.1 | 237.9 | **198.7** | 200.9 | 176.2 | **158.1** | 182.6 | 164.1 | **149.7** |
| | 12 | 310.9 | 294.7 | **262.4** | 212.5 | 198.6 | **177.8** | 174.7 | 168.8 | **159.3** |
| | 18 | 521.4 | 487.4 | **396.7** | 259.3 | 237.9 | **201.8** | 207.1 | 192.4 | **185.8** |
| | 24 | 490.2 | 467.5 | **402.8** | 297.2 | 274.9 | **270.7** | 318.9 | 298.5 | **281.6** |
| t850 | 6 | 1.69 | 1.61 | **1.57** | 1.29 | 1.20 | **1.13** | 1.14 | 1.09 | **1.02** |
| | 12 | 1.79 | 1.67 | **1.61** | 1.55 | **1.48** | 1.49 | 1.24 | 1.18 | **1.11** |
| | 18 | 2.18 | 1.94 | **1.80** | 1.69 | 1.56 | **1.53** | 1.32 | 1.23 | **1.14** |
| | 24 | 2.19 | 2.08 | **1.91** | 1.88 | 1.74 | **1.66** | 1.88 | **1.64** | 1.70 |
| t2m | 6 | 1.65 | 1.58 | **1.48** | 1.76 | 1.68 | **1.59** | 1.49 | 1.43 | **1.36** |
| | 12 | 1.79 | 1.70 | **1.61** | 2.00 | 1.83 | **1.80** | 1.52 | 1.48 | **1.43** |
| | 18 | 2.19 | 2.08 | **1.95** | 2.09 | 2.01 | **1.89** | 1.64 | 1.57 | **1.51** |
| | 24 | 1.99 | 1.94 | **1.86** | 2.14 | 2.05 | **1.99** | 2.05 | **1.98** | 2.00 |
| u10 | 6 | 1.69 | 1.54 | **1.38** | 1.19 | 1.08 | **1.02** | 1.31 | 1.27 | **1.14** |
| | 12 | 2.14 | 2.09 | **2.01** | 1.50 | 1.41 | **1.32** | 1.69 | 1.57 | **1.48** |
| | 18 | 3.15 | 2.94 | **2.76** | 1.75 | 1.54 | **1.38** | 2.00 | 1.91 | **1.84** |
| | 24 | 3.09 | 2.84 | **2.77** | 2.00 | 1.92 | **1.81** | 2.54 | 2.37 | **2.20** |
| v10 | 6 | 1.74 | 1.68 | **1.59** | 1.26 | 1.19 | **1.12** | 1.39 | 1.30 | **1.17** |
| | 12 | 2.33 | 2.18 | **2.04** | 1.55 | 1.39 | **1.30** | 1.69 | 1.55 | **1.49** |
| | 18 | 3.42 | 3.18 | **3.04** | 1.83 | 1.76 | **1.69** | 2.23 | 2.14 | **2.07** |
| | 24 | 3.33 | 3.17 | **3.04** | 2.09 | 1.94 | **1.87** | 2.49 | 2.28 | **2.17** |

**correlation coefficient (ACC)**, indicating the spatial correlations between prediction anomalies $\hat{Y}'$ relative to climatology and ground truth anomalies $Y'$ relative to climatology. **3) Relative Quantile Error (RQE)** (Pathak et al., 2022), which quantifies the relative difference between the model-predicted quantile and the true quantile, indicating whether extremes are overestimated (positive RQE) or underestimated (negative RQE). Detailed formulation for these three metrics can be found in Appendix C.2.

**Baselines.** We utilize ClimaX (Nguyen et al., 2023) as the prediction network, a typical global weather prediction foundation model published by Microsoft. Note that we mainly focus on designing robust loss functions instead of the advancing network architecture. Thus, we compare our Meta-MSE with the other loss functions, including: (1) **RMSE**, the square root of the average of the squared differences between predicted and actual values. (2) **Exloss** (Xu et al., 2024a), proposed a symmetric loss function that assigns larger weights (a fixed constant $\frac{100}{81}$) for extreme weather and keeps normal weights (a fixed constant 1) for typical weather.

**Data preprocessing.** We follow the convention of previous works that normalizes all input data (Nguyen et al., 2023; Meo et al., 2024). For each weather variable, we calculate the mean and standard deviation to normalize them to zero mean and unit variance. In the inference/test phase, we denormalize the output results to the original value range before computing the evaluation metric.

We implement the code and data processing with PyTorch. All experiments were performed on 8 NVIDIA H100 GPUs. To speed up the training process and reduce the use of GPU memory, we also used the FP16 technique. All trials are implemented three times, and the average score is reported. More details about the optimizers, training strategies, and etc can be found in Appendix C.3.

## 5.2 MAIN RESULTS

**Regional Forecasting.** In Table 1, we fine-tune ClimaX model with different losses in three regions, including North America, South America, and Australia, where the data resolution is $5.625°$. Across almost all regions and variables, our proposed DW-MSE consistently achieves the lowest RMSE compared with both the vanilla baseline and the EXloss variant. For example, in z500 forecasting over North America, DW-MSE reduces the 18-hour RMSE from 521.4 (RMSE) and 487.4 (EXloss) to 396.7, a substantial improvement. On near-surface variables such as u10, the advantage is even clearer: in South America at 18 hours, DW-MSE lowers the RMSE to 1.38, compared to 1.75

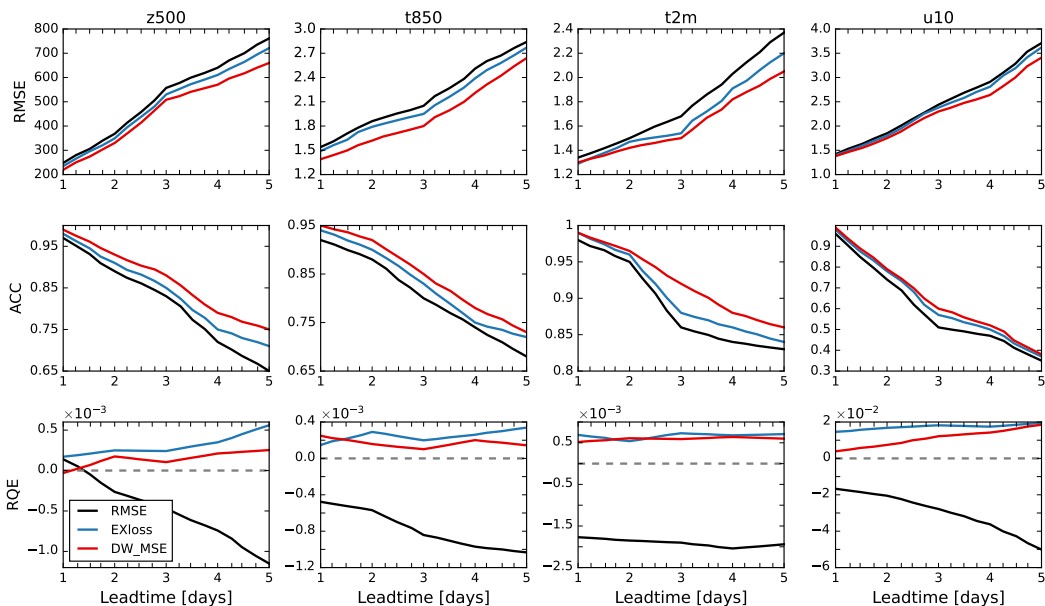

Figure 3: Comparison results with different metrics under *global forecasting*. Note that "RQE < 0" denotes underestimation, and "RQE > 0" denotes overestimation.

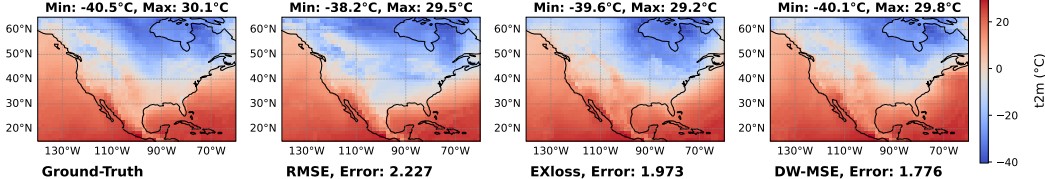

Figure 4: We plot global forecasting results in the region of North America under different losses. Note that *Error* denote the RMSE value computed between prediction results and ground-truth. The value of *t2m* is converted from Kelvin (K) to Celsius (°C).

(RMSE) and 1.54 (EXloss). These consistent improvements across varying settings demonstrate the superiority of DW-MSE.

**Global Forecasting.** In Figure 3, we train a ClimaX model with three different losses from scratch for global forecasting, where the data resolution is $1.40625°$. We report the results of three metrics (RMSE, ACC, and RQE) on five prediction points from Day 1 to Day 5. As the results shown in this table, we observe that our proposed DW-MSE consistently outperforms the baselines across all variables and metrics. For example, in terms of RMSE, DW-MSE achieves lower errors than both RMSE and EXloss, particularly on challenging variables such as t2m, where the Day-5 RMSE is reduced from about 2.3 (RMSE loss) to below 2.1. Similarly, for ACC, DW-MSE maintains higher correlations throughout the forecasting horizon; on z500, the Day-5 ACC of DW-MSE remains above 0.75, compared to only 0.65 for RMSE. Moreover, in terms of RQE, while both EXloss and DW-MSE mitigate the issue of over-smooth prediction (see red/blue lines *vs.* the black line), DW-MSE produces values closer to zero compared to EXloss. It means that our proposed dynamical re-weight mechanism does not lead to significant overfitting on the extreme events.

## 5.3 MORE ANALYSIS

**Visualization.** We plot the results of global forecasting to verify the effectiveness of our proposal on two aspects:

- *Performance in extreme value forecasting.* In Figure 4, we clip the predicted values under different loss functions in the North America region to highlight the local difference in detail. Compared with the baseline RMSE loss, which produces smoothed prediction values and achieves

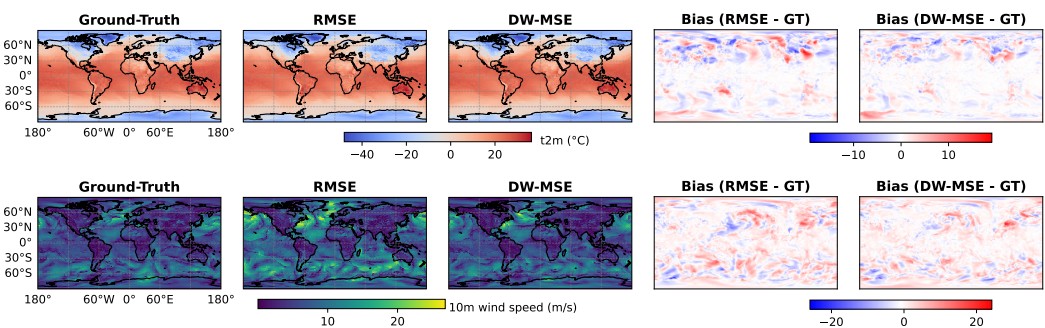

Figure 5: Visualization of global forecasting with the corresponding bias.

Table 2: Effect of each branch in the weighted network. Results for climate variable $z500$ are reported.

| Component | | 72 hours | | 144 hours | |
|:---:|:---:|:---:|:---:|:---:|:---:|
| B1 | B2 | RMSE | ACC | RMSE | ACC |
| | ✓ | 597.3 | 0.764 | 969.3 | 0.541 |
| ✓ | | 562.1 | 0.817 | 842.9 | 0.602 |
| ✓ | ✓ | 508.4 | 0.889 | 782.4 | 0.648 |
| Baseline | | 557.1 | 0.824 | 840.9 | 0.597 |

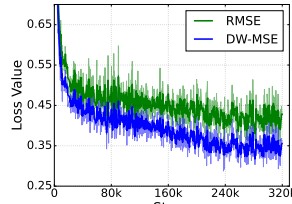

Table 3: Training convergence comparisons with different losses.

higher RMSE values, DW-MSE generates forecasts with less error, with sharper local extreme forecasting (the max and min values are closer to ground truth).

- *Global forecasting performance.* Figure 5 presents the global forecasting results for $t2m$ and *10m wind speed.* Compared with the baseline RMSE loss, DW-MSE produces predictions that are closer to the ground truth, with sharper spatial structures and reduced bias. The bias maps further reveal that DW-MSE effectively suppresses systematic overestimation and underestimation across different regions, particularly in high-latitude and oceanic areas.

These results demonstrate that our reweighting strategy not only improves overall accuracy but also captures local detail variations more accurately, leading to more reliable global forecasts.

**Ablation study.** In Table 2, we evaluate the dual-branch design (B1 for the loss branch and B2 for the weather branch). The baseline refers to training with the standard RMSE loss. Using either B1 or B2 alone already improves performance over the baseline, while combining both achieves the best results—reducing RMSE from 557.1 to 508.4 (72h) and from 840.9 to 782.4 (144h), and consistently boosting ACC. These results highlight the complementary benefits of the two branches.

**Convergence.** Figure 3 illustrates the training loss curves under different loss functions. Compared with the standard RMSE objective, DW-MSE achieves a noticeably faster and more stable convergence. The loss decreases more rapidly in the early training phase and remains consistently lower throughout training, indicating that the proposed reweighting strategy provides more informative gradients for optimization.

## 6 CONCLUSION

In this paper, we propose a robust loss function to address the challenge of overly smooth predictions in climate forecasting. To tackle this issue, we introduce DW-MSE, a meta-learning-based framework that adaptively adjusts weights for MSE losses, allowing the climate prediction network to better capture extreme weather events. Our weighting scheme is neither reliant on prior knowledge nor manually predefined. Through an efficient bilevel optimization strategy that only leverages approximate fires-order information, DW-MSE learns the from both learning dynamics and spatio-temporal climate variables, while enables fewer training costs. Experimental results show that DW-MSE consistently outperforms baseline methods by reducing overall prediction error while improving the accuracy of extreme weather event forecast.

## ETHICS STATEMENT

This work does not present any potential ethical concerns.

## USAGE OF LARGE LANGUAGE MODELS

In this paper, LLMs (like ChatGPT-5, Gemini2.5) are utilized for writing assistance, including grammar correction and text polishing.

## REPRODUCIBILITY STATEMENT

We will release code, configurations, and logs to reproduce all tables and figures, including how to download and reconstruct the benchmarks and the training procedure for regional/global forecasting.

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

## A  METHODOLOGY

### A.1  THE WEIGHTED NETWORK

We formally introduce this network from the following two parts.

**Branch 1, learning from training losses.** In this branch, the input data is training losses represented by $\ell^{\text{train}} \in \mathbb{R}^{H \times W \times V}$. By an operation of "flatten", we can obtain a one-dimensional vector with a dimension of $\mathbb{R}^{1 \times HWV}$, which is subsequently input into a three-layer fully connected MLPs where the architecture is denoted by $\{HWV, \mathrm{W}^{\text{hidden}}, HWV\}$, where $\mathrm{W}^{\text{hidden}}$ is the width of the hidden layer. The output for the first branch denotes

$$\lambda_1 = g_{\boldsymbol{\omega}_1}(\ell^{\text{train}}), \tag{9}$$

where $\boldsymbol{\omega}_1$ denotes the learnable parameters of the first branch. In experiments, we set $\mathrm{W}^{\text{hidden}} = 4HWV$ for convenience.

**Branch 2, learning from weather data.** In the second branch, we introduce a multi-head self-attention (MSA) structure (Voita et al., 2019) to enhance the network's ability to model spatiotemporal dependencies, contributing to the meta-net being aware of extreme cases. Let reshape the climate field $X$ to $\mathrm{Z}_{in} \in \mathbb{R}^{S \times V}$ where $S = H \times W$, which is also the input of one MSA module. The MSA structure uses $B$ heads to jointly attend to the information from different representation sub-spaces. In $b$-th attention head, we first project $\mathrm{Z}_{in}$ to Query $\mathbf{Q}_b$, Key $\mathbf{K}_b$, and Value $\mathbf{V}_b$:

$$\mathbf{Q}_b = \mathrm{Z}_{in}\mathbf{W}_q^b \in \mathbb{R}^{S \times V_k}, \quad \mathbf{K}_b = \mathrm{Z}_{in}\mathbf{W}_k^b \in \mathbb{R}^{S \times V_k}, \quad \mathbf{V}_b = \mathrm{Z}_{in}\mathbf{W}_v^b \in \mathbb{R}^{S \times V_v}, \tag{10}$$

where $\mathbf{W}$ denotes the projection matrices. Then, we can compute the self-attention matrix follows:

$$\mathbf{A}_b = \mathrm{softmax}(\mathbf{Q}_b\mathbf{K}_b^\top / \sqrt{d_k}) \in \mathbb{R}^{S \times S}. \tag{11}$$

This matrix reflects the relationship between tokens, where a larger value in $\mathbf{A}_b$ indicates a stronger interaction between two tokens. The outopt feature of MSA is written as

$$\mathrm{Z}_{out} = \mathrm{MSA}(\mathrm{Z}_{in}) = \mathrm{Concat}([\mathbf{A}_b\mathbf{V}_b]_{b=1}^B)\mathbf{W_0} + \mathrm{Z}_{in}, \tag{12}$$

where $\mathbf{W_0}$ is a projection matrix. Subsequently, we flatten $\mathrm{Z}_{out}$ and input the enhanced feature to a projector (i.e., one layer MLP). At that point, we obtain the generated weight vector from the second branch. For clarity, we formalute the whole process as

$$\lambda_2 = g_{\boldsymbol{\omega}_2}(\mathrm{Z}_{in}) = g_{\boldsymbol{\omega}_2}(X), \tag{13}$$

where $\boldsymbol{\omega}_2$ denotes the learnable parameters of the second branch.

Eventually, we combine the weight vectors from both branches:

$$\lambda = \delta(\gamma\lambda_1 + (1 - \gamma)\lambda_2), \tag{14}$$

where $\delta(\cdot)$ represents the activation function and $\gamma$ is a trade-off hyperparameter. In all experiments, we use the Sigmoid function and set $\gamma = 0.5$.

### A.2  GRADIENT DISCUSSION

Modified by the first-order approximation strategy, the meta-gradient at the training iteration $t$ is converted to

$$\nabla_{\boldsymbol{\omega}}\mathcal{L}^{\text{val}} \approx \frac{\partial \mathcal{L}^{\text{val}}}{\partial f_{\boldsymbol{\theta}}} \cdot \frac{\partial f_{\boldsymbol{\theta}}}{\partial \boldsymbol{\theta}^{(t+1)}}. \tag{15}$$

While this gradient no longer explicitly contains the dependence on the parameters $\boldsymbol{\omega}$ through $\frac{\partial \boldsymbol{\theta}^{(t+1)}}{\partial \boldsymbol{\omega}}$, it can still be effectively updated. This is because that the update of $\boldsymbol{\theta}^{(t+1)}$ in the inner loop still depends on $\boldsymbol{\omega}$ due to the reweighting mechanism in Eq. (5). Therefore, even under only the first-order information, $\boldsymbol{\omega}$ implicitly influences the trajectory of $\theta^{(t+1)}$ via the weighted training loss, which achieves parameters update, while avoiding the prohibitive second-order computation.

### A.3  ALGORITHM

We show the learning algorithm of our proposal DW-MSE in Algorithm 1, where an iterative update process is applied.

---

**Algorithm 1** Learning algorithm of DW-MSE.

---

**Require:** Training dataset $\mathcal{D}^{\mathrm{train}} = \{(X_n, Y_n)\}_{n \in [N]}$, validation dataset $\mathcal{D}^{\mathrm{val}} = \{(X_n, Y_n)\}_{m \in [M]}$, batch size bs, max iterations $T$.
**Ensure:** Prediction network parameters $\theta^{(T)}$.
  1: Initialize $\theta^{(0)}$ for classifier network and $\omega^{(0)}$ for the meta-network.
  2: **for** $t = 1$ **to** $T$ **do**
  3:     Randomly sample two batches $\mathrm{B}_{\mathrm{bs}}^{\mathrm{train}}$ and $\mathrm{B}_{\mathrm{bs}}^{\mathrm{val}}$ from $\mathcal{D}^{\mathrm{train}}$ and $\mathcal{D}^{\mathrm{val}}$, respectively.
  4:     Generate the dynamic weights $[\lambda_1, ..., \lambda_{\mathrm{bs}}]$ for the training samples     ▷ Eq. (4)
  5:     Update $\omega^{(t+1)}$.     ▷ Eq. (5)
  6:     Update $\theta^{(t+1)}$.     ▷ Eq. (8)
  7: **end for**

---

Table 4: List of variables used in the experiments. These two benchmark has the same climate variables, where the only difference is their resolution. "Input" denotes this variable is input for model training and "Output" denotes the prediction model would output this variable.

| Type | Variable name | Abbrev. | Levels | Input | Output |
|------|---------------|---------|--------|-------|--------|
| Static | Land-sea mask | lsm | | ✓ | ✗ |
| Static | Orography | | | ✓ | ✗ |
| Single | 2 metre temperature | t2m | | ✓ | ✓ |
| Single | 10 metre U wind component | u10 | | ✓ | ✓ |
| Single | 10 metre V wind component | v10 | | ✓ | ✓ |
| Atmospheric | Geopotential | z | 500 | ✓ | ✓ |
| Atmospheric | Temperature | t | 850 | ✓ | ✓ |

## B  DATASET

We manually constructed a dataset of ERA5 hourly data on single levels from 1979 to 2018 via the official website by API requests (see `https://cds.climate.copernicus.eu/datasets/`), where the temporal resolution of all data is 6 hours. Then, we regrid these data to two different resolutions, including $5.625°$ and $1.40625°$. More details for the datasets can be found in Table 4 and Table 5.

## C  TRAINING DETAILS

### C.1  TASKS

In experiments, we made comparison with current methods on two tasks, including 1) **Regional forecasting**, we fine-tune a ClimaX model with different loss functions. In this task, the model is required to predict the weather variables within the next 36 hours, where the lead time is 6 hours. Here, we select three representative regions (North America, South America, and Australia), which keeps the same as the setting in previous works Verma et al. (2024); Nguyen et al. (2023). 2) **Global forecasting**, we train a ClimaX model from scratch with different loss functions, where the model is required to generate prediction within the next 144 hours, where the lead time is also 6 hours.

### C.2  METRICS

The formulation for the used three metrics are as follows.

- **Root mean square error (RMSE)**, which measures the average magnitude of prediction errors by taking the square root of the mean of squared differences between predicted and observed values. It is formally written as

$$\mathrm{RMSE} = \sqrt{\frac{1}{UHWV} \sum_{u=1}^{U} \sum_{h=1}^{H} \sum_{w=1}^{W} \sum_{v=1}^{V} \mathcal{W}(z) \big(\mathbf{y}_{u;h,w,v} - \hat{\mathbf{y}}_{u;h,w,v}\big)^2}. \tag{16}$$

Table 5: Details of dataset split.

| Task | resolution | Training | Validation | Test |
|------|-----------|----------|-----------|------|
| Regional forecasting | $5.625°$ | 1990-2015 | 2016 | 2017-2018 |
| Global forecasting | $1.40625°$ | 1979-2015 | 2016 | 2017-2018 |

where $\mathcal{W}(z)$ is the latitude weighting factor that is related to the location in map grids (Nguyen et al., 2023) and $z$ denotes the altitude.

- **Anomaly correlation coefficient (ACC)**, to evaluate the ability of the model to capture anomaly patterns relative to climatology, we compute the anomaly correlation coefficient. Let $\bar{\mathbf{y}}_{h,w,v}$ denote the climatological mean at each grid point. The anomaly fields are defined as

$$\hat{\mathbf{y}}'_{u;h,w,v} = \hat{\mathbf{y}}_{u;h,w,v} - \bar{\mathbf{y}}_{h,w,v}, \quad \mathbf{y}'_{u;h,w,v} = \mathbf{y}_{u;h,w,v} - \bar{\mathbf{y}}_{h,w,v}.$$

Then, the ACC is computed as

$$\text{ACC} = \frac{\sum_{u=1}^{U} \sum_{h=1}^{H} \sum_{w=1}^{W} \sum_{v=1}^{V} \hat{\mathbf{y}}'_{u;h,w,v} \mathbf{y}'_{u;h,w,v}}{\sqrt{\sum_{u=1}^{U} \sum_{h=1}^{H} \sum_{w=1}^{W} \sum_{v=1}^{V} (\hat{\mathbf{y}}'_{u;h,w,v})^2} \cdot \sqrt{\sum_{u=1}^{U} \sum_{h=1}^{H} \sum_{w=1}^{W} \sum_{v=1}^{V} (\mathbf{y}'_{u;h,w,v})^2}}. \tag{17}$$

- **Relative Quantile Error (RQE)** (Pathak et al., 2022), To evaluate the prediction accuracy on extreme events, we compare the quantiles of the predicted and observed fields. Let $Q_p(\hat{\mathbf{y}})$ and $Q_p(\mathbf{y})$ denote the $p$-quantile of the predicted and ground-truth values, computed over all $U \times H \times W \times V$ elements. Then, the RQE at quantile level $p$ is defined as

$$\text{RQE}(p) = \frac{Q_p(\hat{\mathbf{y}}) - Q_p(\mathbf{y})}{Q_p(\mathbf{y})}, \tag{18}$$

when $\text{RQE} > 0$, the extreme value at that quantile is overestimated; when $\text{RQE} < 0$, the extreme value is underestimated.

### C.3 TRAINING STRATEGY

For regional forecasting, we adopt the fine-tuning strategy and load the pre-trained checkpoint from ClimaX model [2]. We use $\text{AdamW}$ optimizer with the combination of parameters $\beta_1 = 0.9, \beta_2 = 0.999$. We set the initial learning rate as $5 \times 10^{-7}$ and the weight decay as $1 \times 10^{-5}$. We set the linear warm-up schedule for the first 5 epochs and a cosine-annealing schedule for the next 45 epochs.

For global forecasting, we train from scratch with a ClimaX model. We use $\text{AdamW}$ optimizer with the combination of parameters $\beta_1 = 0.9, \beta_2 = 0.95$. We set the initial learning rate as $5 \times 10^{-4}$ and the weight decay as $1 \times 10^{-5}$. Similarly, We set the linear warm-up schedule for the first 5 epochs and a cosine-annealing schedule for the next 45 epochs.

For the weighted network, we adopt the Adam optimizer with the initial learning rate of $1 \times 10^{-3}$ and the weight decay of $1 \times 10^{-5}$. Meanwhile, there is no adjustment of the learning rate schedule. We set the batch size as 16 for both two networks.

## D EXPERIMENTS

### D.1 DISCUSSION OF TRAINING OVERHEAD

Compared to existing loss functions such as EXloss, which rely on manually designed and tuned weighting mechanisms, our framework introduces an additional reweighting network for adaptively learning the weights and thus leads to slight computational overhead. Empirically, we compared our proposal with existing loss function in regional forecasting (the North America) and show the results in Table 6. We train the model with 50 epoch in one H100 GPU.

---

[2]https://github.com/microsoft/ClimaX

Table 6: Training cost comparison.

| Metric | Learnable Parameters | GPU memory | Training hours |
|--------|---------------------|------------|----------------|
| RMSE/EXLoss | 107 MB | ∼17.4 GB | 21.7 h |
| Ours | 107 MB + 2.4 MB | ∼21.1 GB | 24.3 h |

Table 7: Performance under different ratio settings for the validation set. Note that "↓" indicates an error reduction and "↑" indicates an error increase relative to the RMSE baseline.

| Ratio | 1 | 1/2 | 1/3 | 1/6 | 1/12 | 1/24 |
|-------|---|-----|-----|-----|------|------|
| z500 | 198.7 (79.4↓) | 199.8 (78.3↓) | 204.8 (73.3↓) | 239.1 (39.0↓) | 257.6 (20.5↓) | 280.4 (2.3↑) |
| t2m | 1.48 (0.17↓) | 1.49 (0.16↓) | 1.52 (0.13↓) | 1.58 (0.07↓) | 1.62 (0.03↓) | 1.67 (0.02↑) |
| t850 | 1.57 (0.12↓) | 1.58 (0.11↓) | 1.62 (0.07↓) | 1.64 (0.05↓) | 1.66 (0.01↑) | 1.69 (0.04↑) |
| u10 | 1.38 (0.31↓) | 1.40 (0.29↓) | 1.43 (0.26↓) | 1.57 (0.12↓) | 1.61 (0.08↓) | 1.68 (0.01↓) |

Our method introduces minimal overhead: the base ClimaX model has about 107 MB of parameters, and the reweighting meta-network adds fewer than 3 MB (under 3% more), while peak GPU memory on an H100 increases only from roughly 17.4 GB to 21.1 GB to store meta-update gradients, and total training time rises modestly from 21.7 h to 25.3 h (roughly 16% overhead), which we believe is justified by the improvements in RMSE and EXLoss, especially for extreme events.

## D.2 IMPACT OF THE SIZE OF VALIDATION SET

In the paper, we follow the setup of ClimaX, where the validation set is constructed using climate data from the year 2016, and testing is performed on data from 2017 and 2018. To further evaluate the robustness of our approach, we conduct additional experiments over the North America region using smaller validation sets — specifically utilizing only $\frac{1}{2}$, $\frac{1}{3}$, $\frac{1}{6}$, $\frac{1}{12}$, or $\frac{1}{24}$ of the 2016 data.

Table 7 reports the next 6-hour prediction errors. The results show that our model remains robust when the validation set is reduced to $\frac{1}{3}$ of its original size, with only mild degradation when using extremely small subsets (e.g., $\frac{1}{12}$ or $\frac{1}{24}$), demonstrating the method's strong generalization ability even under limited validation data.

## D.3 IMPACT OF THE EXTREME THRESHOLD

In our paper, the setting of the extreme value threshold $\rho = 10\%$, which indicates that top/bottom 10% values are selected as the "extreme event". In this section, we explore the sensitivity of this parameter with three settings of the ratio $\rho \in [2\%, 5\%, 10\%]$ for regional forecasting and report the results in the North-America.

As shown in Table 8, we can observe that (1) stable and superior performance. Compared with the baseline, RMSE, our method remains the superior performance across different quantile thresholds. Besdies, the results difference are marginal (within 3% for all climate variables), indicating that the model is not sensitive to the exact choice of the threshold used to define extreme samples. (2) excessively high thresholds (e.g., $\rho = 2\%$) may slightly degrade performance. This is because when the ratio of extreme samples becomes too large, the meta-learning objective puts disproportionate emphasis on rare events, causing the optimization to focus less on the overall distribution. We will further investigate the distribution of the meta-generated weights in future work to better understand how different values of $\rho$ influence the learning dynamics.

Table 8: Performance under different values of $\rho$.

| value of $\rho$ | z500 | | | t2m | | | t850 | | | v10 | | |
|-----------------|------|------|-------|------|------|------|------|------|------|------|------|------|
| | 2% | 5% | 10% | 2% | 5% | 10% | 2% | 5% | 10% | 2% | 5% | 10% |
| Ours | 207.4 | 201.4 | 198.7 | 1.56 | 1.53 | 1.48 | 1.58 | 1.58 | 1.57 | 1.63 | 1.60 | 1.59 |
| RMSE | | 278.1 | | | 1.65 | | | 1.69 | | | 1.74 | |

Table 9: Comparison of NowcastingGPT under different training losses.

| NowcastingGPT | w/ RMSE | w/ EVLoss | w/ Ours |
|---|---|---|---|
| PCC ($\uparrow$) | $0.20 \pm 0.002$ | $0.22 \pm 0.002$ | $\mathbf{0.23} \pm 0.001$ |
| MSE ($\downarrow$) | $3.60 \pm 0.02$ | $3.45 \pm 0.02$ | $\mathbf{3.39} \pm 0.02$ |
| MAE ($\downarrow$) | $0.72 \pm 0.005$ | $0.69 \pm 0.005$ | $\mathbf{0.66} \pm 0.004$ |

## D.4 PRECIPITATION NOWCASTING

In this section, we conduct an experiment on a new task, Precipitation Nowcasting, to verify our method's effectiveness. We refer to the training deatils in the literature (Meo et al., 2024) and adopt the same prediction network, NowcastingGPT. Note that (1) This task is a minute-level climate forecasting, which contains many extreme events, and 2) Different from Climax, NowcastingGPT is a generative-based model. As shown in Table 9, our method consistently improves key metrics over both RMSE and EVLoss, demonstrating that our approach can be integrated with probabilistic or generative forecasters and still yield improvements.

## D.5 VISUALIZATION OF FORECASTING

In Figures 6 and 7, we visualize the ground-truth and prediction results on *South America* region under global forecasting, where the resolution is $1.40625°$. The results verify that our prediction is closer to the real values.

In Figure 8, we visualize the ground-truth, prediction results, and their gap under global forecasting. It clearly shows that DW-MSE achieves close alignment with the ground-truth fields, with the remaining biases exhibiting relatively small magnitude and spatially localized patterns.

Figure 9 illustrates an example of the training sample (t2m) and the corresponding weight distribution produced by our meta-network. It can be observed that regions exhibiting extreme values (e.g., high-latitude areas) are assigned larger weights, while relatively stable regions are assigned smaller weights.

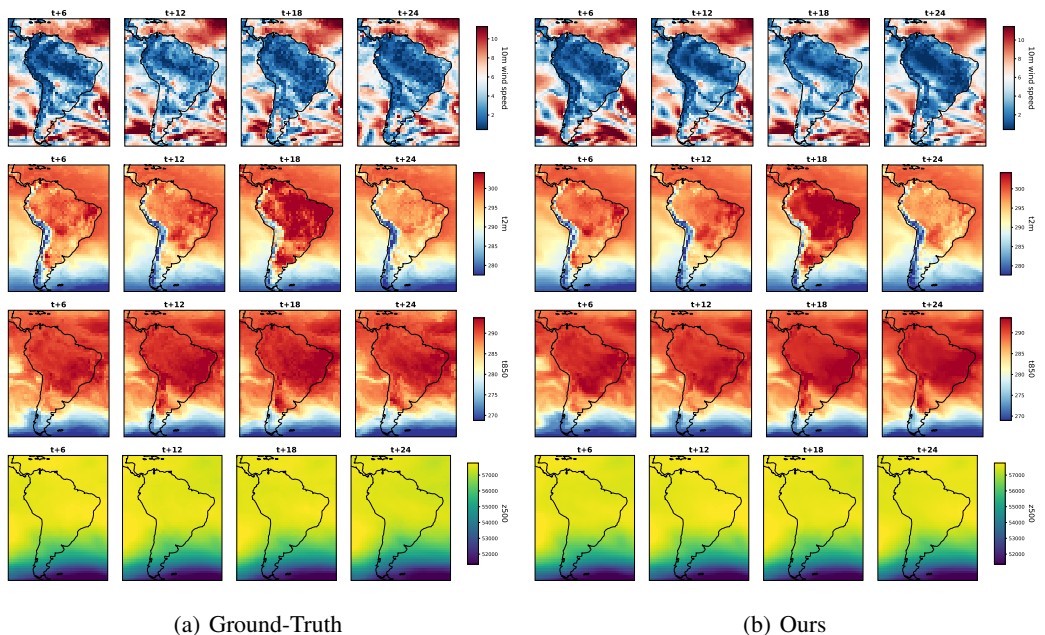

(a) Ground-Truth                                    (b) Ours

Figure 6: Visualization of forecasting results within the next 24 hours on **South America** (lat_range: [-55, 20], lon_range: [270, 330]). Note that these plots is cropped from the results of global forecasting and *10m wind speed* is computed by $\sqrt{u^2 + v^2}$.

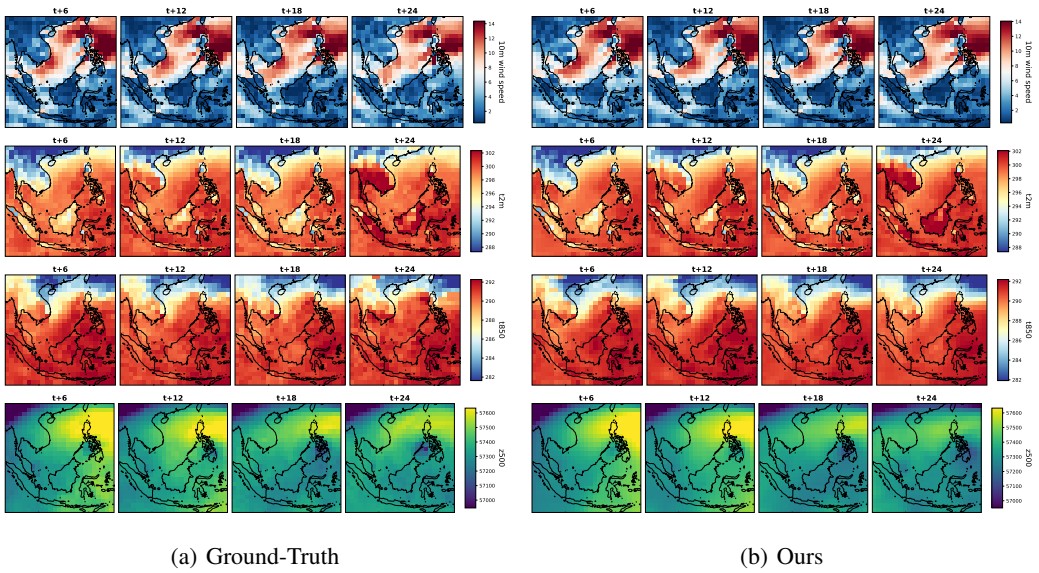

(a) Ground-Truth                                    (b) Ours

Figure 7: Visualization of forecasting results within the next 24 hours on **Southeast Asia** (lat_range: [-10, 25], lon_range: [95, 130]).

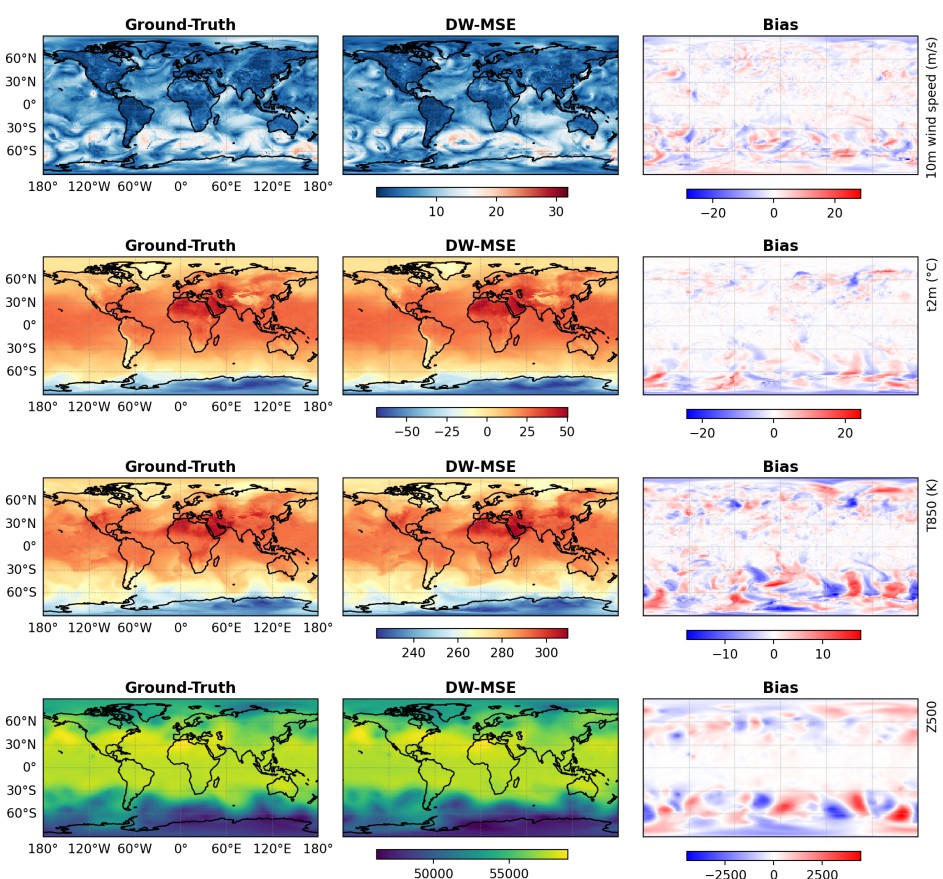

Figure 8: Visualization of prediction bias on four climate variables.

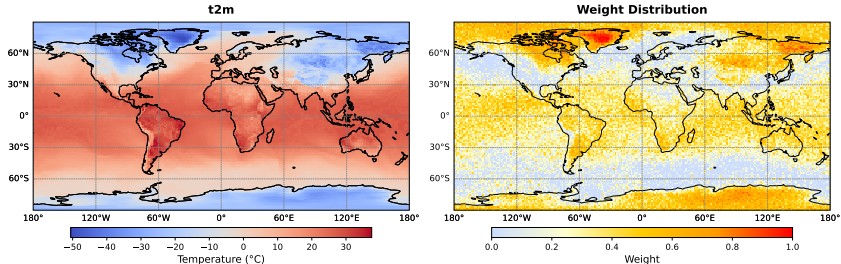

Figure 9: Visualization of training sample and its corresponding generated weight map.

