# OpenReview forum: "Enhancing Extreme Weather Forecasting via Dynamically Weighted MSE"
_ICLR.cc/2026/Conference — Submitted to ICLR 2026_

### Official Review · Reviewer_3BbC · 2025-10-24

**Soundness:** 3
**Presentation:** 2
**Contribution:** 2
**Rating:** 6
**Confidence:** 3

**Summary:**

This paper tackles the challenge that current deep learning weather forecasting models usually produce over-smoothed predictions and perform poorly on extreme weather events. The authors propose a new loss function, Dynamically Weighted MSE (DW-MSE), which introduces a dual-branch meta-network to dynamically assign sample weights, enhancing the learning of extreme samples. One branch of the meta-network captures spatiotemporal relations among climate variables, while the other focuses on training loss information. With an efficient bi-level optimization framework (using an approximate first-order method and avoiding expensive second-order computations), the model learns to up-weight extreme weather samples without manual weighting or human prior. Experiments on both regional and global climate prediction tasks show that DW-MSE surpasses existing loss functions (RMSE, EXloss) in overall accuracy, in predicting extremes, and achieves faster convergence, with more interpretable weighting.

**Strengths:**

The paper proposes a flexible, adaptive loss function which automatically emphasizes extreme weather samples without hand-crafted weights or expert prior, bringing significant improvement in extreme forecasting—a highly practical advantage for real disaster warning applications.

**Weaknesses:**

The “top/bottom percentile” approach for constructing the validation set is under-explained, with no details about the threshold choice, its suitability for all variables/regions, or the risk of bias if extremes are very sparse. In reality, the boundary of what is “extreme” is often ambiguous, making reproducibility and fairness a problem.

Only RMSE and EXloss are included as baselines, which are relatively “basic” weighting schemes. Newer or stronger losses (like EVloss, EVT-based approaches) are missing. Also, the impact of each meta-network branch and bi-level optimization step is not ablated thoroughly (e.g., detailed effects on different types of extreme events), limiting the persuasiveness of claims. In addition, the metrics of ablation experiments should include extreme value metrics.

**Questions:**

How are the top/bottom percentiles chosen? Is there a universal standard across variables/regions? In cases with sparse extremes, how do you ensure the validation set is meaningful and the results are generalizable?

 Can you add comparisons with more diverse/extreme-specific loss functions and provide more thorough ablation (e.g., per-branch effects, per-extreme-type results) to clarify the precise sources of DW-MSE’s advantage?

---

> ### Author Response · Authors · 2025-11-20
>
> **Q1. The selection of top/bottom percentile and its generalization**
>
> Thank you for the thoughtful question. We clarify the percentile choice and how we ensure meaningful evaluation as follows:
>
> - **(1) How the top/bottom percentiles are chosen.** We follow the standard practice used in extreme-event evaluation in weather/climate modeling: for each variable, we identify extremes based on its own value distribution and compute percentiles independently (e.g., top/bottom 5% or 10%). This avoids mixing variables with different physical scales and variances. In this paper, we set this parameter as 10%. Meanwhile, a sensitivity of this parameter can be found in the revised manuscript - Appendix D.3, which shows that our framework is not sensitive to this parameter. Thus, we do not need to carefully select its value via cross-validation.
>
> - **(2) Whether the standard is universal.** While the percentile thresholding strategy is universal, the exact numerical values of the thresholds are computed separately for each variable. This ensures fairness, because what counts as an “extreme” for z500 is very different from that for t2m or u10.
>
> -  **(3) The sparse extreme values.** Because the percentile thresholds are applied within each variable, we do not encounter the issue of extreme-value sparsity—by definition, every field contains exactly the same proportion of top/bottom samples. Our goal here is not to establish a universal domain-wide definition of meteorological extremes, but rather to highlight the model’s ability to better fit the high-impact, high-gradient parts of the distribution. This is precisely the region where traditional MSE-trained models tend to oversmooth.
>
> **Q2. More baselines and more ablation studies**
>
> - **More baselines** Thanks for this constructive suggestion. We agree that including additional baselines would strengthen the evaluation. Here, we add two new methods EVLoss [1] and EPL [2]. Limited to the time restriction, we only conduct experiments in the regional climate forecasting of North America and show the result in the table below. We can see that across all variables and forecast horizons, our method consistently achieves the lowest error, outperforming EVLoss, EXLoss, and EPL by a clear margin and demonstrating its superior ability to enhance extreme and non-extreme forecasting accuracy.
>
> | Variable | Hours | RMSE  | EVloss | EXLoss | EPL | Ours |
> |--|:-:|--:|--:|--:|--:|--:|
> | **z500**  | 6  | 278.1 | 249.2 | 237.9 |224.7| **198.7** |
> |           | 12 | 310.9 | 301.7 | 294.7 |297.1| **262.4** |
> |           | 18 | 521.4 | 479.2 | 487.4 |434.2| **396.7** |
> |           | 24 | 490.2 | 492.4 | 467.5 |441.9| **402.8** |
> | **t850**  | 6  | 1.69  | 1.63 | 1.61  | 1.59 | **1.57** |
> |           | 12 | 1.79  | 1.77 | 1.67  | 1.65 | **1.61** |
> |           | 18 | 2.18  | 2.03 | 1.94  | 1.88 | **1.80** |
> |           | 24 | 2.19  | 2.14 | 2.08  | 2.12 | **1.91** |
> | **t2m**   | 6  | 1.65  | 1.55 | 1.58  | 1.63 | **1.48** |
> |           | 12 | 1.79  | 1.63 | 1.70  | 1.69 | **1.61** |
> |           | 18 | 2.19  | 2.11 | 2.08  | 2.00 | **1.95** |
> |           | 24 | 1.99  | 1.88 | 1.94  | 1.89 | **1.86** |
> | **u10**   | 6  | 1.69  | 1.54 | 1.54  | 1.49 | **1.38** |
> |           | 12 | 2.14  | 2.02 | 2.09  | 2.03 | **2.01** |
> |           | 18 | 3.15  | 2.99 | 2.94  | 2.93 | **2.76** |
> |           | 24 | 3.09  | 3.00 | 2.84  | 3.03 | **2.77** |
>
>
> [1] Extreme Precipitation Nowcasting using Transformer-based Generative Models, ICLR'24
>
> [2] Self-adaptive Extreme Penalized Loss for Imbalanced Time Series Prediction, IJCAI'24
>
> - **More ablation studies on the dual-branch structure** Thanks the reviewer for their insightful comments.  To better evaluate the prediction performance on the extreme-region, we intrudce a new metric, named the Symmetric Extremal Dependence Index (SEDI) - which is commonly used to assess the accuracy of forecasts for extreme climate events. SEDI is computed from the hit rate and false alarm rate, where a higher SEDI value (ranging from 0 to 1) indicates better performance in predicting extreme weather events. The results for ablation studies are shown in the table below. The ablation results show that using B1 (the loss branch) alone already achieves the similar performance compared to the baseline in SEDI across all variables, while combining B1 and B2 achieves the highest SEDI scores. This indicates that the dual-branch design leverages complementary information (training loss and the raw climate data) from the two branches and substantially enhances extreme-event prediction.
>
> | Component |    | Variables |      |      |      |
> |-|-|-|--|-|-|
> | B1        | B2 | z500      | t850 | t2m  | u10  |
> |           | ✓  | 0.80      | 0.79 | 0.81 | 0.73 |
> | ✓         |    | 0.85      | 0.86 | 0.89 | 0.85 |
> | ✓         | ✓  | 0.94      | 0.91 | 0.95 | 0.92 |
> | Baseline  |    | 0.88      | 0.86 | 0.87 | 0.80 |

---

> > ### Comment · Reviewer_3BbC · 2025-11-23
> >
> > Thank you for the authors' response. I believe a score of 6 is appropriate for this paper, and I will therefore maintain my rating.

---

> > > ### Author Response · Authors · 2025-11-26
> > > **Thank you for your feedback**
> > >
> > > We appreciate that you maintain a positive view of our submission. Thank you for your efforts in improving our manuscript.
> > >
> > > Best regards,
> > >
> > > The Authors

---

### Official Review · Reviewer_yEUR · 2025-10-30

**Soundness:** 2
**Presentation:** 2
**Contribution:** 2
**Rating:** 4
**Confidence:** 4

**Summary:**

The paper proposes DW-MSE, a meta-learning loss for data-driven weather forecasting that dynamically reweights training errors to emphasize extremes. A dual-branch “weight network” produces a spatial weight map per sample: one branch ingests the per-grid loss map, the other uses self-attention over raw climate fields. The predictor and weight network are trained with a bilevel scheme using a first-order meta-gradient; the validation signal is computed on masked “extreme-only” grids. Experiments on ERA5 (regional fine-tuning and global training from scratch) report gains over RMSE and ExLoss on RMSE/ACC and reduced RQE bias, with ablations showing both branches help and with faster convergence curves.

**Strengths:**

- Clear motivation and simple interface: The method slots into existing forecasting models and targets a widely observed “over-smoothing” failure mode. The dual-branch design is sensible and well explained.
- Fine-grained, spatial weighting: Producing an H×W×V weight map (rather than a single sample scalar) is a natural refinement for geophysical grids.
- First-order bilevel optimization: Avoids Hessian-vector products yet still improves extremes; the derivation and algorithm are easy to implement.
- Consistent empirical gains in shown experiments

**Weaknesses:**

1.Baselines are too narrow and comparisons are limited to RMSE and EXLoss. A more detailed similar losses in the literature should be discussed, and more baselines should be added to claim state-of-the-art performance.

2.Evaluation scope skews to limited variables. The study focuses on z500, t850, t2m, u10, v10,important but not where extremes most matter operationally (e.g., precipitation, wind gusts, tropical cyclone metrics). A detailed list of experiment results of more variables should be added at least in the appendix.

3.“Extreme-only” validation set design may bias learning. Optimizing the weight network solely on top/bottom ρ-quantiles risks over tuning to tails and neglecting structure near but outside the mask. Please check that non-extreme skill (e.g., mean-state bias) does not degrade.

4.Computational overhead not quantified. The paper argues “efficient” first-order meta-learning, but does not report wall-clock, FLOPs, or GPU memory deltas vs. RMSE/ExLoss. The dual-branch attention over S=H×W tokens could be costly. Please add training time/epoch, total hours, and VRAM for all methods.

5.The ablation study is only done on one variable, please add ablation experiments on more variables to show the comprehensive view of the branches.

**Questions:**

1. The paper cites EVLoss (Meo et al., 2024) but does not include it in experiments. Was it omitted for implementation reasons?
2. The experiments are restricted to five standard variables (z500, t850, t2m, u10, v10). Have the authors evaluated DW-MSE on precipitation, surface pressure, or wind gusts where extreme behavior is most pronounced?
3. How sensitive are the results to ρ, and does the non-extreme skill remain stable?
4. Could the authors report wall-clock training time, FLOPs, or memory usage versus RMSE and ExLoss to substantiate the efficiency claim?
5. Table 2 reports ablations only on z500. Did similar patterns hold for other variables?
If the author can address my concerns I am happy to raise my score.

---

> ### Author Response · Authors · 2025-11-20
> **Response to Reviewer yEUR (1/2)**
>
> **Q1&W1: More Baselines**
>
> Thanks for this constructive suggestion. We agree that including additional baselines would strengthen the evaluation. Here, we add two new methods EVLoss [1] and EPL [2]. Limited to the time restriction, we only conduct experiments in the reginoal climate forecasting of North America and show the result in the table below. We can see that across all variables and forecast horizons, our method consistently achieves the lowest error, outperforming EVLoss, EXLoss, and EPL by a clear margin and demonstrating its superior ability to enhance extreme and non-extreme forecasting accuracy.
>
> | Variable | Hours | RMSE  | EVloss | EXLoss | EPL | Ours |
> |-----------|:------:|------:|-------:|-------:|-------:|-----------:|
> | **z500**  | 6  | 278.1 | 249.2 | 237.9 |224.7| **198.7** |
> |           | 12 | 310.9 | 301.7 | 294.7 |297.1| **262.4** |
> |           | 18 | 521.4 | 479.2 | 487.4 |434.2| **396.7** |
> |           | 24 | 490.2 | 492.4 | 467.5 |441.9| **402.8** |
> | **t850**  | 6  | 1.69  | 1.63 | 1.61  | 1.59 | **1.57** |
> |           | 12 | 1.79  | 1.77 | 1.67  | 1.65 | **1.61** |
> |           | 18 | 2.18  | 2.03 | 1.94  | 1.88 | **1.80** |
> |           | 24 | 2.19  | 2.14 | 2.08  | 2.12 | **1.91** |
> | **t2m**   | 6  | 1.65  | 1.55 | 1.58  | 1.63 | **1.48** |
> |           | 12 | 1.79  | 1.63 | 1.70  | 1.69 | **1.61** |
> |           | 18 | 2.19  | 2.11 | 2.08  | 2.00 | **1.95** |
> |           | 24 | 1.99  | 1.88 | 1.94  | 1.89 | **1.86** |
> | **u10**   | 6  | 1.69  | 1.54 | 1.54  | 1.49 | **1.38** |
> |           | 12 | 2.14  | 2.02 | 2.09  | 2.03 | **2.01** |
> |           | 18 | 3.15  | 2.99 | 2.94  | 2.93 | **2.76** |
> |           | 24 | 3.09  | 3.00 | 2.84  | 3.03 | **2.77** |
>
> [1] Extreme Precipitation Nowcasting using Transformer-based Generative Models, ICLR'24
>
> [2] Self-adaptive Extreme Penalized Loss for Imbalanced Time Series Prediction, IJCAI'24
>
>
> **Q2&W2: More climate Variables**
>
> Thanks for the reviewer’s insightful comment. Firstly, we would like to clarify that our study focuses on five commonly used variables (z500, t850, t2m, u10, and v10), which are standard evaluation variables in recent large-scale climate forecasting benchmarks such as ClimaX and ClimODE. These variables effectively represent key atmospheric layers and dynamics (from geopotential height to near-surface wind and temperature) and are thus widely adopted for evaluating model performance. Besides, **our forecasting lead time ranges from 6 to 24 hours (hour-level prediction), which may not be optimal for variables such as precipitation that evolve primarily at minute or sub-hourly scales.**
>
> Secondly, to further examine the generalizability of our proposed framework beyond these variables, we **conduct a trial on minute-level precipitation forecasting.** Following the setting in the literature [1], we adopted the NowcastingGPT generative model, which originally incorporates an extreme-value-aware objective (EVLoss). The results are summarized below:
>
> | NowcastingGPT | w/ RMSE | w/ EVLoss  | w/ Ours |
> |---------------|---------|-----------|---------|
> | PCC ($\uparrow$)      |  0.20 $\pm$ 0.002 | 0.22 $\pm$ 0.002 | **0.23** $\pm$ 0.001 |
> | MSE ($\downarrow$)    |  3.60 $\pm$ 0.02  | 3.45 $\pm$ 0.02  | **3.39** $\pm$ 0.02  |
> | MAE ($\downarrow$)    |  0.72 $\pm$ 0.005 | 0.69 $\pm$ 0.005 | **0.66** $\pm$ 0.004 |
>
> These results demonstrate that our dual-weighted meta-learning framework can also improve the accuracy and stability of precipitation prediction, even when applied to short-term, highly dynamic variables.

---

> ### Author Response · Authors · 2025-11-20
> **Response to Reviewer yEUR (2/2)**
>
> **Q3&W3: (1) Sensitivity to $\rho$**
>
> Here, we explore the sensitivity of this parameter with three settings of the ratio $ \rho \in [2\%, 5\%, 10\%]$ for regional forecasting and report the results in the North-America as the following table.
> |      | z500 |    |     | t2m |    |     | t850 |    |     | v10 |    |     |
> |------|------|----|-----|-----|----|-----|------|----|-----|-----|----|-----|
> | RMSE |278.1 |    |     |1.65 |    |     | 1.69 |    |     | 1.74|    |     |
> |Ratio $\rho$ | 2%   | 5% | 10% | 2%  | 5% | 10% | 2%   | 5% | 10% | 2%  | 5% | 10% |
> | Ours | 207.4 | 201.4 | 198.7| 1.56 | 1.53 |1.48| 1.60 | 1.58 |1.57 | 1.63  | 1.60 |1.59 |
>
> We can observe that (1) stable and superior performance. Compared with the baseline, RMSE, our method remains the superior performance across different quantile thresholds. Besdies, the results difference are marginal (within 3% for all climate variables), indicating that the model is not sensitive to the exact choice of the threshold used to define extreme samples. (2) excessively high thresholds (e.g., $\rho = 2\%$) may slightly degrade performance. This is because when the ratio of extreme samples becomes too large, the meta-learning objective puts disproportionate emphasis on rare events, causing the optimization to focus less on the overall distribution. We will further investigate the distribution of the meta-generated weights in future work to better understand how different values of $\rho$ influence the learning dynamics.
>
> **Q3&W3: (2) Performance on non-extreme region**
>
> Thanks for this insightful comment. Here, we conduct an extra trival that separates the performance of the prediction model on extreme/non-extreme regions and show them in the table below. Our method substantially improves performance in both extreme and non-extreme regions, demonstrating our proposal's effectiveness.
>
> |        | Extreme |     |      |     | Non-Extreme |     |      |     |
> |--------|---------|-----|------|-----|-------------|-----|------|-----|
> |        | z500    | t2m | t850 | u10 | z500        | t2m | t850 | u10 |
> | RMSE   | 305.1 | 1.70 | 1.71 | 1.72 | 264.8  | 1.61 |1.63 | 1.63 |
> | Ours   | 200.4 | 1.52 | 1.60 | 1.42 | 184.2  | 1.46| 1.52 |1.34|
>
> **Q4&W4: Training cost**
>
> Thanks for this insight comment and we will add this discussion to the paper. Compared to existing loss functions such as EXloss, which rely on manually designed and tuned weighting mechanisms, our framework introduces an additional reweighting network for adaptively learning the weights and thus leads to slight computational overhead.  Empirically, we compared our proposal with existing loss function in regional forecasting and show the results in the following table. We train the model with 50 epoch in one H100 GPU. The batch size is 16.
>
> | Metric | Learnable Parameters  | GPU memory | Total training hours |
> |---:|---:|---:|---:|
> | RMSE/EXLoss   | 107MB          | ~17.4GB | 21.7h |
> | Ours          | 107MB + 2.4 MB | ~21.1GB | 25.3h |
>
> Our method introduces minimal overhead: the base ClimaX model has about 107 MB of parameters, and the reweighting meta-network adds fewer than 3 MB (under 3% more), while peak GPU memory on an H100 increases only from roughly 17.4 GB to 21.1 GB to store meta-update gradients, and total training time rises modestly from 21.7 h to 25.3 h (≈16% overhead), which we believe is justified by the improvements in RMSE and EXLoss, especially for extreme events.
>
> Besides, **the efficiency of our method mainly comes from the first-order approximation to the bilevel optimization.** If we were to adopt a traditional bilevel optimization algorithm with second-order gradients, the training would require more than 110 GPU hours. In contrast, our first-order formulation reduces the cost to under 30 hours.
>
> **Q5&W5: Extension of Table 2 (Results for more variables)**
>
> We extend the prediction results of the ablation study in the table below, where the prediction leadtime is 72 hours.  We can see that enabling only one branch provides moderate gains over the baseline, but activating both B1 and B2 consistently achieves the lowest RMSE and the highest ACC for all variables. This demonstrates that the two branches are complementary and jointly essential for capturing complex spatial patterns and improving extreme-weather prediction accuracy.
>
> | Component |      | z500 |        | t850 |        | t2m | | u10| |
> |-|-|-|-|-|-|-|-|-|-|
> | B1       | B2    | RMSE     | ACC   | RMSE      | ACC  |RMSE      | ACC    |RMSE      | ACC    |
> |          | ✓     | 597.3    | 0.76  | 3.39     | 0.69  |3.01      | 0.71   |3.71      | 0.49    |
> | ✓        |       | 562.1    | 0.81  | 3.14     | 0.76  |2.62      | 0.79   |3.42      | 0.54    |
> | ✓        | ✓     | 508.4    | 0.88  | 2.61     | 0.85  |2.47      | 0.86   |3.20      | 0.69    |
> |   Baseline   |   | 557.1    | 0.82  | 3.27     | 0.74  |2.74      | 0.82   |3.64      | 0.51    |

---

> ### Author Response · Authors · 2025-11-26
> **Looking forward to your feedback**
>
> Dear Reviewer yEUR,
>
> Thanks for your valuable review comments. We understand that you might be extremely busy at this time, thus we would deeply appreciate it if you could take some time to provide further feedback on whether our rebuttal solves your concerns. Kindly let us know if our response has adequately addressed your concerns.
>
> Best Regards,
>
> The Authors

---

> > ### Comment · Reviewer_yEUR · 2025-11-28
> > **Response to Author**
> >
> > Dear authors,
> >   Thank you for your additional experiments and clarification, they address all my concerns, but it seems that currently reviewers are unable to change scores, I will check back later.

---

> > > ### Author Response · Authors · 2025-11-29
> > > **Thank you for your feedback**
> > >
> > > Dear reviewer,
> > >
> > > Thank you very much for your follow-up and for letting us know that the additional experiments and clarifications have addressed your concerns. We truly appreciate the time and effort you have devoted to our paper.
> > >
> > > By the way, we wonder If you feel that our work now warrants a more positive assessment. We would be very grateful if you could consider adjusting your score or recommendation, as this may be important for the final decision.
> > >
> > > Thank you again for your thoughtful and constructive review.
> > >
> > > Best regards,
> > >
> > > All authors

---

### Official Review · Reviewer_EvRG · 2025-11-01

**Soundness:** 3
**Presentation:** 3
**Contribution:** 2
**Rating:** 4
**Confidence:** 4

**Summary:**

The paper proposes a weighted MSE scheme to improve the accuracy of extreme weather forecasts, particularly by addressing the oversmoothing issue in weather models.

**Strengths:**

The paper clearly explains a common problem in weather prediction and offers a simple solution. The set-up is explained in an easy to understand manner.

**Weaknesses:**

- It doesn't become clear to me why the setup makes the network aware of extremes, the weights are according to forecast error, which might in some cases correspond to extreme events, but could also just be difficult ot predict terrain or something completely different.
- It would be interesting to show how many data points need to be in the validation set for the method to work correctly.
- While the error metrics presented are interesting, there is such a large emphasis on the over-smoothing problem in the motivation of the paper, that I would have expected metrics related to smoothness or at least a stronger focus on the extremes (there's only some average quantile error information in the appendix and one analysis).
- I like that multiple areas of the world are considered, but the three selected are still a bit of an odd sub-sample, which isn't discussed enough (like e.g. what would you expect in other areas of the world).
- I believe some form of skill score would be better for Figure 5. It's hard to see any differences in these plots between the methods.

**Questions:**

Why does the setup make the network aware of extremes? The weights are based on forecast error, which might in some cases correspond to extreme events, but could also reflect difficult-to-predict terrain or something completely different.

---

> ### Author Response · Authors · 2025-11-20
> **Response to Reviewer EvRG (1/2)**
>
> **Q1&W1: Why this dynamically reweighting mechanism works?**
>
> We appreciate the reviewer’s thoughtful question. Indeed, forecast errors can arise from various sources, not only extreme events but also regions with complex dynamics such as mountainous terrain or coastal boundaries. **Our design does not only rely on instantaneous error to define “extremes”. Instead, the meta-network learns context-aware reweighting patterns based on a dual-branch design which contains both the prediction error and the associated physical variables.** Through this dual-branch design, one branch encodes the forecast error features, while the other captures the climatological context (e.g., variable type, spatial region, and seasonality). This allows the meta-network to differentiate genuine extreme events from structurally difficult regions, emphasizing samples that are both rare and impactful rather than merely high-error.
>
> **W2: Sensitivity of the number of validation set**
>
> Thanks for this interesting suggestion. In the paper, we follow the setup of ClimaX, where the validation set is constructed using climate data from the year 2016, and testing is performed on data from 2017 and 2018. To further evaluate the robustness of our approach, we conduct additional experiments over the North America region using smaller validation sets — specifically utilizing only $\frac{1}{2}$, $\frac{1}{3}$, $\frac{1}{6}$, $\frac{1}{12}$, or $\frac{1}{24}$ of the 2016 data.
>
> | Ratio | 1 | 1/2 | 1/3 | 1/6 | 1/12 | 1/24 |
> |-------|---|-----|-----|-----|------|------|
> | z500  | 198.7(79.4$\downarrow$)|199.8(78.3$\downarrow$) |204.8(73.3$\downarrow$)|239.1(39.0$\downarrow$)| 257.6(20.5$\downarrow$)|280.4(2.3$\uparrow$)|
> | t2m   |1.48(0.17$\downarrow$)|1.49(0.16$\downarrow$)|1.52(0.13$\downarrow$)|1.58(0.07$\downarrow$)|1.62(0.03$\downarrow$)|1.67(0.02$\uparrow$)|
> | t850  |1.57(0.12$\downarrow$)|1.58(0.11$\downarrow$)|1.62(0.07$\downarrow$)|1.64(0.05$\downarrow$)|1.66(0.01$\uparrow$)|1.69(0.04$\uparrow$)|
> | u10   |1.38(0.31$\downarrow$)|1.40(0.29$\downarrow$)|1.43(0.26$\downarrow$)|1.57(0.12$\downarrow$)|1.61(0.08$\downarrow$)|1.68(0.01$\downarrow$)|
>
> This table reports the next 6-hour prediction errors, where “$\downarrow$” indicates an error reduction and “$\uparrow$” indicates an error increase relative to the RMSE baseline. The results show that our model remains robust when the validation set is reduced to $\frac{1}{3}$ of its original size, with only mild degradation when using extremely small subsets (e.g., $\frac{1}{12}$ or $\frac{1}{24}$), demonstrating the method’s strong generalization ability even under limited validation data.
>
>
> **W3: More Metrics on extreme-value prediction**
>
> Thanks for the reviewer’s insightful comment. Firstly, we would like to emphasize thta the metric RQE in Figure 3 is utilized to evaluate the prediction accuracy on extreme events (positive/negative value denotes over-/under- estimation for the extrem value). We can see that our method achieves more accurate prediction compared with two baselines.
>
> Here, we also introduce an additional evaluation metric - **the Symmetric Extremal Dependence Index (SEDI)** - which is commonly used to assess the accuracy of forecasts for extreme climate events. SEDI is computed from the hit rate and false alarm rate, where a higher SEDI value (ranging from 0 to 1) indicates better performance in predicting extreme weather events.
>
>
> | Leadtime | 1 day |      | 2 day |      | 3 day |      |
> |----------|-------|------|-------|------|-------|------|
> |          | t2m   | wind | t2m   | wind | t2m   | wind |
> | RMSE     | 0.87  | 0.88 | 0.83  | 0.81 | 0.79  | 0.78 |
> | EXloss   | 0.91  | 0.91 | 0.88  | 0.87 | 0.85  | 0.80 |
> | Ours     | 0.95  | 0.92 | 0.90  | 0.89 | 0.86  | 0.83 |
>
> In this table, we report results for two variables, t2m and wind, where wind is calculated as $\sqrt{u_{10}^2 + v_{10}^2}$. As shown, our method consistently achieves higher SEDI scores across all lead times, indicating stronger extremal dependence and superior skill in capturing extreme weather behaviors compared to RMSE and EXLoss-based baselines. We will incorporate this additional experiment and analysis into the revised version of the paper to provide a more comprehensive evaluation of model performance under extreme conditions.

---

> ### Author Response · Authors · 2025-11-20
> **Response to Reviewer EvRG (2/2)**
>
> **W4: Consideration about more regions**
>
> We appreciate the reviewer’s thoughtful comment. The three selected regions—North America, South America, and Australia—were chosen for two main reasons.
>
> 1) First, they represent diverse yet complementary climate regimes across both hemispheres. North and South America span a wide range of latitudes, covering tropical, subtropical, and temperate zones, which provide a relatively uniform distribution of climatic variability. In contrast, Australia exhibits more stable yet distinct meteorological characteristics, with persistently warm and dry inland conditions and strong coastal variability. This combination allows us to evaluate model performance under both highly dynamic and relatively stable climate conditions.
>
> 2) Second, the results of these regions are reported in the some papers (like ClimaX and ClimODE), which makes them ideal for fair and direct comparison.
>
> In the later version of our paper, we plan to extend our experiments to additional regions such as Southeast Asia, which frequently experiences strong precipitation events. We aim to construct a minute-level precipitation dataset to further evaluate our framework’s effectiveness in early warning and prediction of intense precipitation.
>
>
> **W5: Format of Figure 5**
>
> We thank the reviewer for this helpful suggestion. We agree that global-scale visualizations may be unclear to demonstrate the superiority of our method. In future work, we plan to crop and analyze specific regional subdomains (e.g., coastal areas) to better highlight the framework's improved ability to capture and fit extreme events. These additional visualizations will be included in the revised version to provide a clearer comparison between methods.

---

> ### Author Response · Authors · 2025-11-26
> **Looking forward to your feedback**
>
> Dear Reviewer EvRG,
>
> Thanks for your valuable review comments. We understand that you might be extremely busy at this time, thus we would deeply appreciate it if you could take some time to provide further feedback on whether our rebuttal solves your concerns. Kindly let us know if our response has adequately addressed your concerns.
>
> Best Regards,
>
> The Authors

---

### Official Review · Reviewer_e9dc · 2025-11-02

**Soundness:** 3
**Presentation:** 3
**Contribution:** 3
**Rating:** 4
**Confidence:** 4

**Summary:**

Current deep learning weather models, while powerful, are hindered by standard loss functions like RMSE that produce oversmoothed forecasts and fail to accurately predict critical extreme weather events.

To address this, the authors propose DW-MSE (Dynamically Weighted MSE), a novel meta-learning framework that uses a dual-branch network to automatically and adaptively assign higher importance to extreme weather samples during training. This eliminates the need for manual, pre-defined weighting schemes used in prior work.

The method is optimized efficiently with a bi-level strategy and is shown through extensive experiments to consistently improve extreme weather forecasting performance, outperforming existing approaches.

**Strengths:**

1) It directly addresses a well-known and significant weakness in current deep learning weather models—their failure to accurately forecast high-impact extreme weather events due to oversmoothing from standard loss functions.

2) The core strength is its move away from manually designed, rigid weighting functions. The dual-branch meta-network automatically learns how to assign importance to samples, making it more flexible and potentially more effective than prior heuristic approaches.

3) The framework does not require extensive pre-defined knowledge about what constitutes an "extreme event" in the data, allowing it to discover and adapt to complex patterns inherently.

**Weaknesses:**

1) The proposed framework is significantly more complex than a standard model with a simple MSE loss. It requires training and coordinating two networks (the prediction network and the dual-branch meta-network) instead of one, which could make implementation and debugging more challenging. The meta-network's optimization is "guided by a small set of validation samples." The performance of the entire system could be sensitive to how this validation set is constructed and whether it is truly representative of the extreme events the model needs to learn.

2) While the goal is to improve extreme event prediction, a potential risk is that the model might overfit to these events at the cost of degrading performance for more common, "normal" weather patterns. The introduction states it maintains "strong overall forecasting performance," but this is a key trade-off to monitor.

3) The experiments contain no baselines, training cost and hyper-parameter search report.

**Questions:**

See weakness.

---

> ### Author Response · Authors · 2025-11-20
> **Response to Reviewer e9dc (1/2)**
>
> **W1: (1) Complexity of our framework; (2) Construction of the validation set and its sensitivity**
>
> Thanks for the reviewer's insightful comments. We split this weaknesses into three parts and give the corresponding response as the follows
> 1) **The complexity of our framework**. Firstly, we acknowledge that our framework involves an additional reweighting network compared to other methods. However, the training process is simple (end-to-end) and does not introduce significant computation costs (only involve first-order information), while not involving various hyper-parameters compared to those mannually-design reweighting loss functions (like EXloss and EVloss). We believe that these advantages, together with the strong empirical performance, demonstrate its value in this research area.
>
> 2) **How to build this validation set**. The construction of validation set is not complex. In practical implementation, the extreme samples are defined as the top and bottom 10% of values across all climate variables. During model training, the validation set is constructed using the same quantile thresholds to ensure consistency between the training and validation distributions.
>
> 3) **Sensitivity of the validation set**. Here, we conduct experiments to verify its sensitivity from two aspects, including
>
> i) **Number of the validation set samples.** In the paper, we follow the setup of ClimaX, where the validation set is constructed using climate data from the year 2016, and testing is performed on data from 2017 and 2018. To further evaluate the robustness of our approach, we conduct additional experiments over the North America region using smaller validation sets — specifically utilizing only $\frac{1}{2}$, $\frac{1}{3}$, $\frac{1}{6}$, $\frac{1}{12}$, or $\frac{1}{24}$ of the 2016 data.  This table reports the next 6-hour prediction errors, where “$\downarrow$” indicates an error reduction and “$\uparrow$” indicates an error increase relative to the RMSE baseline. The results show that our model remains robust when the validation set is reduced to $\frac{1}{3}$ of its original size, with only mild degradation when using extremely small subsets (e.g., $\frac{1}{12}$ or $\frac{1}{24}$), demonstrating the method’s strong generalization ability even under limited validation data.
> | Ratio | 1 | 1/2 | 1/3 | 1/6 | 1/12 | 1/24 |
> |-------|---|-----|-----|-----|------|------|
> | z500  | 198.7(79.4$\downarrow$)|199.8(78.3$\downarrow$) |204.8(73.3$\downarrow$)|239.1(39.0$\downarrow$)| 257.6(20.5$\downarrow$)|280.4(2.3$\uparrow$)|
> | t2m   |1.48(0.17$\downarrow$)|1.49(0.16$\downarrow$)|1.52(0.13$\downarrow$)|1.58(0.07$\downarrow$)|1.62(0.03$\downarrow$)|1.67(0.02$\uparrow$)|
> | t850  |1.57(0.12$\downarrow$)|1.58(0.11$\downarrow$)|1.62(0.07$\downarrow$)|1.64(0.05$\downarrow$)|1.66(0.01$\uparrow$)|1.69(0.04$\uparrow$)|
> | u10   |1.38(0.31$\downarrow$)|1.40(0.29$\downarrow$)|1.43(0.26$\downarrow$)|1.57(0.12$\downarrow$)|1.61(0.08$\downarrow$)|1.68(0.01$\downarrow$)|
>
> ii) **The value of $\rho$**. We adjust the gradient singal from the meta-net by controling the value of $\rho$ in the validation set. Here, we explore the sensitivity of this parameter with three settings of the ratio $ \rho \in $[2%, 5%, 10%] for regional forecasting and report the results in the North-America as the following table.
>
> |      | z500 |    |     | t2m |    |     | t850 |    |     | v10 |    |     |
> |------|------|----|-----|-----|----|-----|------|----|-----|-----|----|-----|
> | RMSE |278.1 |    |     |1.65 |    |     | 1.69 |    |     | 1.74|    |     |
> |Ratio $\rho$ | 2%   | 5% | 10% | 2%  | 5% | 10% | 2%   | 5% | 10% | 2%  | 5% | 10% |
> | Ours | 207.4 | 201.4 | 198.7| 1.56 | 1.53 |1.48| 1.58 | 1.58 |1.57 | 1.63  | 1.60 |1.59 |
>
> We can observe that (1) stable and superior performance. Compared with the baseline, RMSE, our method remains the superior performance across different quantile thresholds. Besdies, the results difference are marginal (within 3% for all climate variables), indicating that the model is not sensitive to the exact choice of the threshold used to define extreme samples. (2) excessively high thresholds (e.g., $\rho$=2%) may slightly degrade performance. This is because when the ratio of extreme samples becomes too large, the meta-learning objective puts disproportionate emphasis on rare events, causing the optimization to focus less on the overall distribution. We will further investigate the distribution of the meta-generated weights in future work to better understand how different values of $\rho$ influence the learning dynamics.

---

> ### Author Response · Authors · 2025-11-20
> **Response to Reviewer e9dc (2/2)**
>
> **W2: The learning objective and trade-off issue**
>
> We appreciate the reviewer’s insightful comment. The concern regarding potential overfitting to extreme events is reasonable. In our design, the dual-branch meta-network does not exclusively optimize for extremes but instead balances the gradient contributions between common and rare samples through meta-weighting guided by a small validation set. When the model produces smooth predictions, the training objective yields smaller gradients; in contrast, large gradients (typically corresponding to extreme events) prompt the meta-network to adaptively re-scale their weights. This allows the model to assign greater importance to extreme cases without distorting the overall learning distribution.
>
> Empirically, as shown in Figure 3, our model maintains comparable or even better performance in global forecasting while achieving substantial gains on extreme-event metrics. In summary, this dynamic re-scaling mechanism effectively enhances robustness to extremes without sacrificing general performance.
>
> Here, we conduct an extra trival that separates the performance of the prediction model on extreme/non-extreme regions and show them in the table below. Our method substantially improves performance in both extreme and non-extreme regions, demonstrating our proposal's effectiveness.
>
> |        | Extreme |     |      |     | Non-Extreme |     |      |     |
> |--------|---------|-----|------|-----|-------------|-----|------|-----|
> |        | z500    | t2m | t850 | u10 | z500        | t2m | t850 | u10 |
> | RMSE   | 305.1 | 1.70 | 1.71 | 1.72 | 264.8  | 1.61 |1.63 | 1.63 |
> | Ours   | 200.4 | 1.52 | 1.60 | 1.42 | 184.2  | 1.46| 1.52 |1.34|
>
> **W3: Baseline, Training Cost, and Hyper-parameters**
>
> i) **Baseline**. Our framework aims to propose a stronger reweighting strategy (instead of designing a better network architecture like Pangu-Weather, FuXi, and etc) to tackle the smooth-prediction issue in extreme climate prediction. Therefore, we compared with the naive training baseline (RMSE) and existing reweight loss function (Exloss). In both global and reginoal forecasting with sevrious metrics, our method consistently outperform their performane.
>
> ii) **Training Cost**. Although our framework consists of two networks, the training cost does not increase significantly. In this table, we compared our proposal with existing loss function in regional forecasting and show the results in the following table. We train the model with 50 epoch in one H100 GPU.
>
> | Metric | Learnable Parameters  | GPU memory | Training hours |
> |---|---:|---:|---:|
> | RMSE/EXLoss   | 107MB          | ~ 17.4GB | 21.7h |
> | Ours          | 107MB + 2.4 MB | ~ 21.1GB | 25.3h |
>
> Our method introduces minimal overhead: the base ClimaX model has about 107 MB of parameters, and the reweighting meta-network adds fewer than 3 MB (under 3% more), while peak GPU memory on an H100 increases only from roughly 17.4 GB to 21.1 GB to store meta-update gradients, and total training time rises modestly from 21.7 h to 25.3 h (≈16% overhead), which we believe is justified by the improvements in RMSE and EXLoss, especially for extreme events.
>
> iii) **Hyper-parameters**.
> Thanks for this constructive suggestion. In practice, our adaptive reweighting mechanism involves fewer hyperparameters compared to manually designed reweighting methods such as EXLoss and EVLoss. In fact, there is only one key hyperparameter—the trade-off weight $\lambda$ between the dual branches—which controls the balance between the learning signal derived from the training loss and that from the raw climate data.
>
> Here, we made ablation studies where $\lambda \in [0.1, 0.3, 0.5, 0.7, 0.9]$. Limited to the page restriction, we only reported the results of two variables in the following table. From the table, we observe that the model performance remains highly stable across a wide range of trade-off weights $\lambda \in [0.1, 0.9]$, which indicates that the method maintains robust performance without requiring precise hyperparameter tuning.
>
> |      | z500  |     |       |     |     | t2m  |     |      |     |     |
> |------|-------|-----|-------|-----|-----|------|-----|------|-----|-----|
> |**RMSE** | 278.1 |     |       |     |     | 1.65 |     |      |     |     |
> |  **Value of $\lambda$**    | 0.1   | 0.3 | 0.5   | 0.7 | 0.9 | 0.1  | 0.3 | 0.5  | 0.7 | 0.9 |
> |**Ours** | 202.5 | 198.7 | 198.7 | **196.8** | 200.6 |1.50 |1.49 | **1.48** | 1.49 |1.49|

---

> ### Author Response · Authors · 2025-11-26
> **Looking forward to your feedback**
>
> Dear Reviewer e9dc,
>
> Thanks for your valuable review comments. We understand that you might be extremely busy at this time, thus we would deeply appreciate it if you could take some time to provide further feedback on whether our rebuttal solves your concerns. Kindly let us know if our response has adequately addressed your concerns.
>
> Best Regards,
>
> The Authors

---

### Official Review · Reviewer_QPAQ · 2025-11-07

**Soundness:** 3
**Presentation:** 3
**Contribution:** 2
**Rating:** 6
**Confidence:** 3

**Summary:**

The paper introduces a new method called dynamically weighted mean squared error (DW-MSE) to improve weather forecasting, particularly for extreme events. Traditional deep learning models often produce overly smooth predictions because they minimize average errors, which causes them to miss rare but severe events. DW-MSE addresses this by automatically assigning greater importance to extreme weather samples during training through a dual-branch system that learns from both prediction errors and the weather data itself. This adaptive weighting process is optimized efficiently without relying on manual adjustments or prior knowledge. Experiments show that DW-MSE improves both regional and global forecasts, capturing sharper and more accurate extreme weather patterns while maintaining strong overall performance.

**Strengths:**

- The paper is in general well written and nicely structured.
- The paper convincingly shows that MSE-trained models smooth out extremes and miss high-impact events.
- The authors present a simple but effective idea, that is, to learn a spatial weight map for MSE instead of hand-crafted weights, so it plugs into any forecaster models with minimal code changes.
- Clear indication that the dual-branch meta-net (using both loss maps and raw fields with self-attention) is best (both braches helps and together is the best).
- First-order update avoids Hessians and keeps the method effectively trainable.
- Bias maps and regional cutouts make the "less smoothing, sharper extremes" claim easy to verify.
- Methods shows faster and steadier training than plain MSE.

**Weaknesses:**

- The results seemingly rely on an "extremes-only" validation set. The quantile threshold is under-specified and sensitivity is not analyzed.
- The theoretical contribution is somewhat limited. There are no stability or convergence guarantees for the learned weights. The first-order choice is justified empirically only.
- Baselines seem to be quite narrow. There are only few comparisons to recent probabilistic or generative approaches aimed at extremes.
- Compute cost is not reported (GPU-hours, memory, wall-time) despite the extra outer loop. Hence it remains unclear how it would scale.
- Possible trade-off against non-extreme performance is not quantified beyond RQE.

**Questions:**

- What quantile threshold defines the extremes in the validation set, and how sensitive are results to it across variables and regions?
- What is the training overhead vs. RMSE and EXLoss (GPU-hours, peak memory, wall-time)?
- Can this be combined with probabilistic forecasters or ensembles, and does it improve spread–skill?
- You may add histogram- or log-binned residual analyses to show the distribution of error magnitudes, not just means. Include per-quantile curves and spatial error maps to reveal where large errors concentrate.

---

> ### Author Response · Authors · 2025-11-20
>
> **Q1&W1: Setup of the extreme threshold $\rho$ and its sensitivity**
>
> We apologize for the unclear statement about the setup of the extreme threshold. In practice, extreme samples are defined as the top and bottom 10% of values across all climate variables, and the validation set uses the same quantile thresholds to keep the training and validation distribution consistent.
>
> Here, we explore the sensitivity of this parameter with three settings of the ratio $ \rho \in$ [2%, 5%, 10%] for regional forecasting and report the results in the North-America as the following table.
> | Variables | z500 |    |     | t2m |    |     | t850 |    |     | v10 |    |     |
> |-|-|-|-|-|-|-|-|-|-|-|-|--|
> | RMSE | 278.1 |    |     |1.65 |    |     | 1.69 |    |     | 1.74|    |     |
> |Ratio $\rho$ | 2%   | 5% | 10% | 2%  | 5% | 10% | 2%   | 5% | 10% | 2%  | 5% | 10% |
> | Ours | 207.4 | 201.4 | 198.7| 1.56 | 1.53 |1.48| 1.60 | 1.58 |1.57 | 1.63  | 1.60 |1.59 |
>
>
> We can observe that (1) stable and superior performance. Compared with the baseline, RMSE, our method remains the superior performance across different quantile thresholds. Besdies, the results difference are marginal (within 3% for all climate variables), indicating that the model is not sensitive to the exact choice of the threshold used to define extreme samples. (2) excessively high thresholds (e.g., $\rho$=2%) may slightly degrade performance. This is because when the ratio of extreme samples becomes too large, the meta-learning objective puts disproportionate emphasis on rare events, causing the optimization to focus less on the overall distribution.
>
> **W2: The theoretical contribution**
>
> Thanks for the insightful comment. We agree that providing stability or convergence guarantees for first-order approximation strategy will largely enhance the contribution of our method. Later, we will try to follow a line of influential meta-optimization works to provide the theoretical analysis for our work.
>
> **Q2: Training cost**
>
> Thanks for your constructive comments. Compared to existing loss functions such as EXloss, which rely on manually designed and tuned weighting mechanisms, our framework introduces an additional reweighting network for adaptively learning the weights and thus leads to slight computational overhead.
>
> Empirically, we compared our proposal with existing loss function in regional forecasting and show the results in the following table. We train the model with 50 epoch in one H100 GPU.
>
> | Metric | Learnable Parameters  | GPU memory | Training hours |
> |-:|-:|-:|-:|
> | RMSE/EXLoss   | 107MB          | ~17.4GB | 21.7h |
> | Ours          | 107MB + 2.4 MB | ~21.1GB | 25.3h |
>
> Our method introduces minimal overhead: the base ClimaX model has about 107 MB of parameters, and the reweighting meta-network adds fewer than 3 MB (under 3% more), while peak GPU memory on an H100 increases only from roughly 17.4 GB to 21.1 GB to store meta-update gradients, and total training time rises modestly from 21.7 h to 25.3 h (≈16% overhead), which we believe is justified by the improvements in RMSE and EXLoss, especially for extreme events.
>
> **Q3&W3: Comparison with generative-based method**
>
> Our framework is not restricted to any specific architecture of the prediction network. To examine its applicability, we conduct experiments here to test our method on NowcastingGPT [1], a generative (probabilistic) forecasting network. As shown below, our method consistently improves key metrics over both RMSE and EVLoss, demonstrating that our approach can be integrated with probabilistic or generative forecasters and still yield improvements.
>
> | NowcastingGPT | w/ RMSE | w/ EVLoss  | w/ Ours |
> |-|-|-|-|
> | PCC ($\uparrow$)      |  0.20 $\pm$ 0.002 | 0.22 $\pm$ 0.002 | **0.23** $\pm$ 0.001 |
> | MSE ($\downarrow$)    |  3.60 $\pm$ 0.02  | 3.45 $\pm$ 0.02  | **3.39** $\pm$ 0.02  |
> | MAE ($\downarrow$)    |  0.72 $\pm$ 0.005 | 0.69 $\pm$ 0.005 | **0.66** $\pm$ 0.004 |
>
> [1] Extreme Precipitation Nowcasting using Transformer-based Generative Models, ICLR'24
>
> **Q4: Distribution of error magnitudes in the plot**
>
> Thanks for this constructive suggestion. In the later version, we will revise and add the prediction variance among different trivals for the main table and change the plot to a histogram style.
>
> **W5: More results in both extreme and non-extreme region**
>
> Thanks for this insightful comments. Here, we separate the performance of the prediction model on extreme/non-extreme regions (in North America) and show the 6-hour prediction result in the table below.
>
> |        | Extreme |     |      |     | Non-Extreme |     |      |     |
> |-|-|-|-|-|-|-|-|-|
> |        | z500    | t2m | t850 | u10 | z500        | t2m | t850 | u10 |
> | RMSE   | 305.1 | 1.70 | 1.71 | 1.72 | 264.8  | 1.61 |1.63 | 1.63 |
> | Ours   | 200.4 | 1.52 | 1.60 | 1.42 | 184.2  | 1.46| 1.52 |1.34|
>
> Our method substantially improves performance in both extreme and non-extreme regions, demonstrating our proposal's effectiveness

---

### Author Response · Authors · 2025-11-20
**Summary of our rebuttal**

We sincerely thank all reviewers for their thoughtful comments and constructive suggestions. We are grateful for the positive assessments regarding the motivation, clarity, and effectiveness of our approach.


Below, we summarize the main concerns raised by the reviewers along with our corresponding solutions. Based on these review comments, we also modify our manuscript (highlighted with blue font).
1) **Training Cost** (Reviewer **QPAQ, e9dc, yEUR**). We provided the training cost comparison including training time, GPU memory utilization, and the number of network parameters. This part has been added to the **Appendix D.1**.
2) **Construction of the validation set and its impacts** (Reviewer **QPAQ, e9dc, EvRG**), We explained the construction details of the validation set and explored the impact of the number of validation sample. This part can be found in **AppendiX D.2**.
3) **Setup of the parameter $\rho$ and its sensitivity** (Reviewer **QPAQ, e9dc, 3BbC**). We provided a detailed setting of this parameter and conducted experiments to verify its sensitivity. This part has been added to the **Appendix D.3**.

Due to time constraints, we’re unable to revise the manuscript during the rebuttal phase in response to every reviewer comment, but we’ll use the feedback to further improve the quality of our work afterward.

---

### Author Response · Authors · 2025-12-03
**Summary of our paper, the reviewer's comments, and our reponse (Part 1/2)**

Dear AC,

Due to the changes in rebuttal stage, we are here to provide a brief summary of our work and our responses to the reviewers' comments, which might helps AC's final decision.

### **Summary of our work:**

- **Task:** We propose a novel loss function, Dynamically Weighted Mean Squared Error (DW-MSE), designed to enhance the forecasting accuracy of deep learning models specifically for extreme weather events.

- **Motivation and Novelty:** We observe that while standard loss functions (i.e., MSE) often lead to "smoothing" of predictions and underestimation of rare, high-impact events, our DW-MSE addresses this by automatically assigning higher loss weight to large prediction errors that correlate with the missed extreme weather magnitudes. This provides a dynamic focusing mechanism, unlike static weighting schemes.

- **Contribution:**
  1) **Adaptive Reweighting Loss Function:** We introduce a novel loss function built on a meta-learning framework that adaptively reweights training samples via a dual-branch meta-network.
  2) **Efficient Optimization Algorithm:** We propose an approximate first-order optimization algorithm which significantly reduces the training cost compared to traditional bi-level optimization methods.
  3) **Performance Improvement:** Our method consistently achieves robust performance improvements across multiple climate forecasting settings compared to existing extreme-value target loss functions.


### **Summary of review's comments and our responses:**

**Strengths:**

- **Importance and Motivation of the Problem:**

  Reviewer **QPAQ**, **e9dc**, **EvRG**, **yEUR** and **3BbC** all emphasized that the paper tackles a well-known and practically important weakness of current deep learning weather models—oversmoothing and poor performance on extreme events. **QPAQ** and **yEUR** noted that the method directly targets extreme‐event forecasting while maintaining overall performance, and **e9dc** highlighted that it addresses a “significant weakness” in existing systems.

- **Simplicity, Practicality, and Applicability of the Approach:**

  Reviewers **QPAQ**, **EvRG**, **yEUR** and **3BbC** described the proposed DW-MSE as a simple but effective solution. **EvRG** and **yEUR** noted that the set-up is easy to understand and slots naturally into existing forecasting models, while **3BbC** stressed that the loss automatically emphasizes extreme samples without hand-crafted weights or expert prior, making it highly practical for real disaster-warning applications.

- **Writing and structure of the paper:**

  Reviewers **QPAQ**, **EvRG** and **yEUR** remarked that the paper is well written and clearly structured. **EvRG** specifically stated that the paper clearly explains the common problem and offers a simple solution, and **yEUR** noted that the motivation and interface of the method are clear and well explained.

---

### Author Response · Authors · 2025-12-03
**Summary of our paper, the reviewer's comments, and our reponse (Part 2/2)**

**Negative Reviewers and our response**

- Response to Reviewer **e9dc (score: 4)**
  1) **Complexity of our framework:** The training process of our proposeal is simple (end-to-end) and does not introduce significant computation costs (only involve first-order information), while not involving various hyper-parameters compared to those mannually-design reweighting loss functions (like EXloss and EVloss).
  2) **Construction of the validation set and the impact of its sample number:** We add content to our manuscript which clearly states the construction manner and conduct more ablation studies to verify the sensitivity of sample numbers.
  3) **The learning objective and trade-off issue:** We add some experiments to show that our method can improve the performance of prediction model on both extreme and non-extreme value regions.
  4) **Training Cost, and Hyper-parameters:** We add more experiments to show that our method will not significantly introduce additional computation cost and conduct ablation studies to verify the impact of one unessential hyperparameter.

- Response to Reviewer **EvRG (score: 4)**
  1) **Why our method works:** We give the intuitive explanation for our motivation and the method.
  2) **Sensitivity of the number of validation set:** We add content to our manuscript which clearly states the construction manner and conduct more ablation studies to verify the sensitivity of sample numbers.
  3) **More Metrics on extreme-value prediction:** We add one metric that focuses on the performance of prediction model on extreme-value region. The results show the superiority of our proposal compared with existing methods.
  4) **Consideration about more regions:** In the paper, we conduct experiments of regional forecasting in North America, South America, and Australia. In the later version of our paper, we plan to extend our experiments to additional regions such as Southeast Asia, which frequently experiences strong precipitation events.
  5) **The format of one figure in the paper:** In future work, we plan to crop and analyze specific regional subdomains (e.g., coastal areas) to better highlight the framework's improved ability to capture and fit extreme events.

- Response to Reviewer **yEUR (score: 4), (Note: they stated that they are willing to raise the socre in previous rebuttal)**
  1) **More baselines:** We add two additional robust loss functions, EVloss and EPL and make more experiments. The results also demonstrate our method's superiority.
  2) **More climate Variables:** we conduct a trial on minute-level precipitation forecasting, where the results show that our method improves the accuracy and stability of precipitation prediction, even when applied to short-term, highly dynamic variables.
  3) **Sensitivity to $\rho$:** We add experiments to test the sensitivity of this parameters.
  4) **Performance on non-extreme region:** We split the extreme and non-extreme region and show the corresponding performance, which also demonstrates our proposal's effectiveness.
  5) **Training cost:** We provide more comparison results on training time, GPU memory, and the number of learnable parameters.
  6) **Extension of Table 2:** We provide more results.


Thank you for your time and consideration.

Best regards,

The Authors

---

### Meta-Review · Area_Chair_76hE · 2026-01-07

**Summary:**

This paper presents an algorithm that improves existing weather forecasting models for extreme weather forecasting with a new loss function, DW-MSE.

The reviewers raised the following weaknesses.
* The proposed algorithm was tested only on small-scale and low-resolution data.
* There is a risk of weakening the weather forecasting performance on normal regions.
* There are insufficient baseline methods.
* Only 5 variables have been tested (the AC notes: this is reasonable).

**Reviewer Concerns:**

The authors tried to address the above concerns, but many of them cannot be addressed, such as the low-resolution nature and the limited comparison against strong models. Overall, the AC believes that the paper has not well-validated the effectiveness of the proposed algorithm.

**Reviewer Scores:**

I think all reviewers will arrive at a consensus of 4, weak rejection.

---

### Decision · Program_Chairs · 2026-01-26

Reject